# Extravascular gelation shrinkage-derived internal stress enables tumor starvation therapy with suppressed metastasis and recurrence

Kun Zhang [1,2]*, Yan Fang[1], Yaping He[1], Haohao Yin[1], Xin Guan[1], Yinying Pu[1], Bangguo Zhou[1], Wenwen Yue[1], Weiwei Ren[1], Dou Du[1], Hongyan Li[1], Chang Liu[1], Liping Sun[1], Yu Chen [3]* & Huixiong Xu[1]*

Despite the efficacy of current starvation therapies, they are often associated with some intrinsic drawbacks such as poor persistence, facile tumor metastasis and recurrence. Herein, we establish an extravascular gelation shrinkage-derived internal stress strategy for squeezing and narrowing blood vessels, occluding blood & nutrition supply, reducing vascular density, inducing hypoxia and apoptosis and eventually realizing starvation therapy of malignancies. To this end, a biocompatible composite hydrogel consisting of gold nanorods (GNRs) and thermal-sensitive hydrogel mixture was engineered, wherein GRNs can strengthen the structural property of hydrogel mixture and enable robust gelation shrinkage-induced internal stresses. Systematic experiments demonstrate that this starvation therapy can suppress the growths of PANC-1 pancreatic cancer and 4T1 breast cancer. More significantly, this starvation strategy can suppress tumor metastasis and tumor recurrence via reducing vascular density and blood supply and occluding tumor migration passages, which thus provides a promising avenue to comprehensive cancer therapy.

[1] Department of Medical Ultrasound, Shanghai Tenth People's Hospital, and Ultrasound Research and Education Institute, Tongji University School of Medicine, Tongji University, 301 Yan-chang-zhong Road, Shanghai 200072, P. R. China. [2] National Center for International Research of Bio-targeting Theranostics, Guangxi Key Laboratory of Bio-targeting Theranostics, Collaborative Innovation Center for Tumor-targeting Theranostics, Guangxi Medical University, 22 Shuang-Yong Road, Nanning, Guangxi 530021, P. R. China. [3] State Key Laboratory of High Performance Ceramics and Superfine Microstructure, Shanghai Institute of Ceramics, Chinese Academy of Sciences, Shanghai 200050, P. R. China. *email: zhang1986kun@126.com; chenyu@mail.sic.ac.cn; xuhuixiong@126.com

Progression, recurrence, and metastasis of malignant tumor are three primary challenges that current therapeutic protocols are confronted with[1,2]. In an attempt to address these challenges[3,4], starvation therapy has aroused considerable interest, because it can inhibit tumor via decreasing nutrient supply that is indispensible for tumor growth[5,6]. Rapid ingredient consumption (i.e., oxygen scavenging, glucose exhaustion, and nutrition intake occlusion) and blood vessel occlusion are two main approaches to give rise to nutrition deficiency and realize starvation therapy[7–9]. In general, rapid ingredient consumption inevitably occurs to normal cells or tissues and this process is not persistent, since the ever-existing and intact blood supply will continuously provide nutrition. In contrast, blood vessel occlusion strategy capable of permanently occluding blood & nutrition supply is preferred, among which two typical protocols, i.e., intratumoral vascular rupture and intravascular aggregate-induced embolism, are highlighted[10–12]. Various science communities have witnessed the advance of this blood vessel occlusion-based starvation therapy[10,11]. However, the two protocols in blood vessel occlusion strategy also confront some intractable challenges, i.e., tumor metastasis, tumor recurrence, and normal blood vessel embolism. More significantly, in the intravascular aggregate-induced embolism protocol, rapid blood flow will wash away the initial nuclei and hamper agglomerate formation, probably disabling vascular occlusion. Fortunately, the in-depth insight into starvation therapy inspires us to pursue new protocols capable of addressing these challenges.

As a versatile biomaterial, hydrogels have garnered considerable attention[12–15], and they have exhibited significant potential as scaffolds or matrix for drug delivery, cell incubation, and implants[16–23]. However, as one fundamental structure-determining factor, the ubiquitous internal stress in hydrogel gelation is still regarded as an ominous sign and even as a severe hazard endangering mechanical properties of hydrogels[19–22,24]. As a result, most research focused on how to eliminate internal stress rather than exploring its potential value. Actually, from the opposite perspective, internal stresses during gelation also displayed several certain usages, e.g., drug release, micro-/super-lenses and complex three-dimensional (3D) structures[20,25,26].

Enlightened by them, we, in this report, established another starvation therapy protocol, namely extravascular gelation shrinkage-derived internal stress for squeezing blood vessels, cutting off blood and nutrition supply essential for tumor growth and ultimately inhibiting tumor growth. To exemplify it, an organic/inorganic hybrid composite containing PEG-SH-modified gold nanorods (GNR-PEG-SH) and composite hydrogel (chitosan (CS)/mPEG-Mal/pNIPAAm-co-AAc) is constructed (hydrogel–GNR). With PANC-1 pancreatic tumor and 4T1 breast tumor as models, hydrogel–GNR irradiated by 808 nm laser can produce internal stress during GNR-mediated gelation shrinkage, squeeze and narrow intratumoral and adjoining blood vessels and cut off nutrition supply, consequently inhibiting tumor progression on both PANC-1 and 4T1 solid tumors. Moreover, this starvation therapy can considerably suppress the spontaneous lung metastasis from primary tumor at breast of transgenic mouse model via reducing vascular density, cutting off blood and nutrition supply and occluding migration passages[27–29]. Furthermore, it also impedes recurrence and metastasis of highly invasive PANC-1 pancreatic cancer after two special physical therapies (i.e., photo-thermal ablation and ultrasound mechanical destruction), which thus provides another choice for comprehensive cancer therapy. Moreover, this well-established starvation therapy deriving from extravascular gelation shrinkage-derived internal stress can address the two problematic issues that current intravascular aggregate-induced embolism protocols encounter, i.e., initial nuclei leaving-caused failure of agglomerate generation and normal blood vessel embolism.

## Results

**Synthesis and characterization of hydrogel–GNR.** GNR-mediated gelation of hydrogel–GNR is shown in Fig. 1a, wherein GNRs can mediate the photothermal transition, trigger gelation and volume shrinkage upon exposure to 808 nm laser irradiation[30] and produce internal stresses. In principle, four pivotal parameters, i.e., low critical solution temperature (LCST), porosity, toughness, and stiffness associated with composition and structure characteristics decide the degree of internal stresses[21,22,31]. LCST between 37 °C and 42 °C is appropriate and beneficial for evaluating this starvation therapy via removing the interference of hyperpyrexia-induced apoptosis[32]. Fortunately, pNIPAAm-co-AAc hydrogels serve as the primary framework and determine the LCST of hydrogel–GNR, because the LCST of pNIPAAm-co-AAc can be adjusted via tuning AAc ratio according to the practical demand[33], that is, more hydrophobic segments will obtain larger LCST[34,35]. In this regard, the LCST of pNIPAAm-co-AAc can be elevated to 38.7 °C from 32 °C corresponding to pure pNIPAAm (Fig. 1b) when AAc segment percentage in pNIPAAm-co-AAc make ups approximately 20% (Fig. 1c)[36]. In particular, the introductions of mPEG-Mal and CS significantly reduce porosity and pore diameter via the physical cross-linking (Fig. 1f)[37,38], undoubtedly benefiting the reinforced compactness and internal stress.

Thiol-terminated PEG-modified GNRs (i.e., GNR-PEG-SH) with a length/diameter ratio of 6:1 were employed to mediate the gelation process (Supplementary Fig. 1) due to their typical near-infrared absorption peak at 808 nm (Supplementary Fig. 2)[39]. GNRs can be conjugated with PEG in the composite hydrogel (CS/mPEG-Mal/pNIPAAm-co-AAc) via the spontaneous reaction (Fig. 1d)[39], and their the maximum absorption wavelength fails to be altered during gelation (Supplementary Fig. 3). Depending on their strong photothermal transition at 808 nm (Supplementary Figs 4 and 5), GNRs successfully mediate the sol–gel transition process of hydrogel–GNR (50 ppm) (CS/mPEG-Mal/pNIPAAm-co-AAc/GNRs) (Fig. 1e). Besides attaining photothermal-triggered gelation, GNRs can further reduce porosity and pore size of the ultimate hydrogel–GNR through weaving its 3D spatial network and augmenting cross-linking joints (Fig. 1f)[40,41]. Therefore, GNRs introduction is expected to contribute to the reinforced mechanical strength and storage modulus of the composite hydrogels, favorably augmenting the gelation shrinkage-derived internal stress. This phenomenon is analogous to previous reports using nanoparticles mingling or composition tuning for enhancing mechanical properties of hydrogels[41–44].

Liquid hydrogel–GNR manifests a Newton-fluid property, ensuring its syringeability available for in vivo application (Supplementary Fig. 6). Noticeably, GNRs introduction fails to alter LCST of the ultimate product, since the abrupt inflection points of transmission rate and viscosity remain at 38.7 °C (Fig. 2a and Supplementary Fig. 7). To further comprehend the influence of GNRs on gelation temperature, hydrogel–GNR was subjected to a variable temperature rheological sweep. The crossover between $G'$ and $G''$ that marks gelation formation further corroborates the preferable transition temperature at approximately 39 °C (Supplementary Fig. 8).

**Mechanical and rheological analysis.** To demonstrate GNRs conjugation-mediated improvements of mechanical and rheological properties, various dynamic measurements were carried out. Compressive stress–strain inspection indicates that contributed by the significantly augmented weaving and junction point density by GNRs conjugation[42], hydrogel–GNR can withstand larger strain than hydrogel alone and display an anti-compression

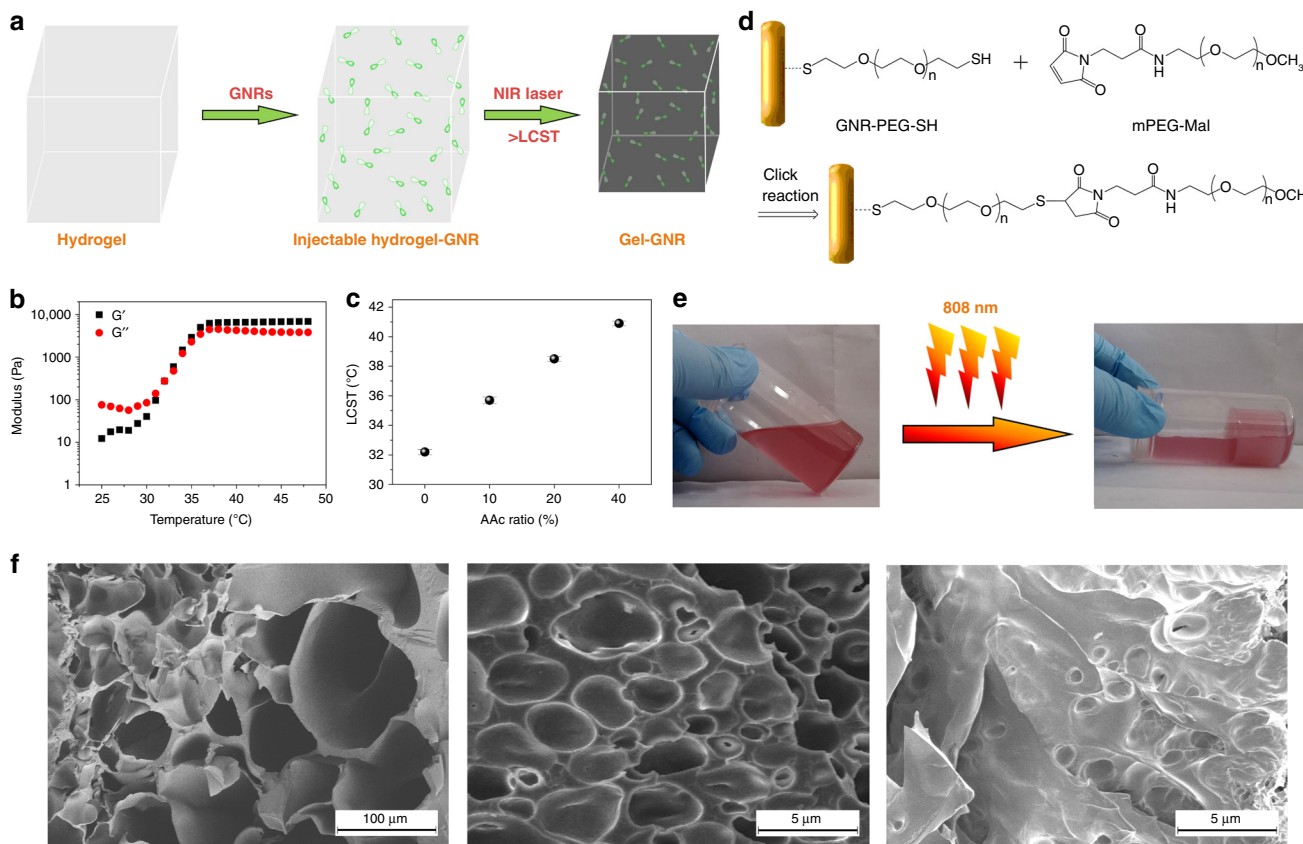

**Fig. 1** Synthesis and characterizations of hydrogel–GNR. **a** Schematic on the gelation process of hydrogel–GNR. **b** Temperature-dependent rheological curves of pNIPAAm without AAc segment, wherein the crossover between $G'$ and $G''$ that represent storage modulus and loss modulus, respectively, is determined as the gelation temperature (that is, LCST). **c** LCST of pNIPAAm-*co*-AAc with tunable AAc content (0%, 10%, 20%, and 40%) determined by dynamic oscillatory temperature sweeps, and data are expressed as mean ± standard deviation (SD) ($n = 3$). **d** Schematic of chelation reaction between GNR-PEG-SH and mPEG-Mal in CS/mPEG-Mal/pNIPAAm-*co*-AAc. **e** Digital photos of hydrogel–GNR (50 ppm) before and after 808 nm irradiation, and 20% AAc in pNIPAAm-*co*-AAc was employed. **f** Scanning electron microscopic (SEM) images of pNIPAAm-*co*-AAc (left), composite hydrogel (CS/mPEG-Mal/pNIPAAm-*co*-AAc) (middle), and hydrogel–GNR (right) with 20 ppm GNRs, wherein 20% AAc in pNIPAAm-*co*-AAc was employed.

behavior, suggesting the increased toughness. More significantly, GNRs chelation substantially increases the failure stress beyond which fracture occurs (Fig. 2b), which suggests that hydrogel–GNR can endure larger stresses. This impressive phenomenon imparts hydrogel–GNR with larger strength and elastic modulus arising from increased mechanical stiffness (Supplementary Fig. 9)[36]. As a result, the hydrogel–GNR holds considerable potential in resisting blood pressure and squeezing blood vessels of tumor via the gelation shrinkage-derived internal stress. Noticeably, excessive and unchelated GNRs that can impede the cross-linking of components play a negative effect on the mechanical properties of hydrogel–GNR when comparing 50 ppm to 20 ppm. In the following experiments, hydrogel–GNR specifically refers to hydrogel–GNR (20 ppm) unless indicated otherwise.

Afterwards, strain-dependent oscillatory rheology was measured to assess the mechanical stiffness of different samples. Network failure fails to occur until strain is stretched to over 70%, which is in accordance with aforementioned result. Higher yielding point of hydrogel–GNR than hydrogel indicates that the hydrogel–GNR can withstand larger deformation and accommodate larger toughness. Notably, before reaching the yielding point, much larger $G'$ of hydrogel–GNR suggests that GNRs chelation can significantly enhance the mechanical stiffness of hydrogel–GNR by augmenting multivalent joints (Fig. 2c). Furthermore, $G'$ and $G''$ as a function of the angular frequency were determined. The mechanical properties are dominated by $G'$

across the range of frequencies, exhibiting a great anti-shear ability and high strength (Fig. 2d). In particular, GNRs introduction indeed influences the rheological properties of hydrogel–GNR via improving the mechanical stability. Time-dependent rheological result demonstrates shorter gelation time when introducing GNRs (Fig. 2e, f), which is beneficial for rapid solidification when applied in vivo.

**Ex vivo and in vivo vascular narrowing and occlusion test.**
Depending on these impressive improvements in mechanical stiffness and reduced porosity and pore size, it is no doubt that such an extravascular gelation shrinkage-induced internal stress will occlude blood & nutrition supply via squeezing blood vessels and resisting blood pressure. To verify it, ex vivo blood vessels were utilized to explore how this internal stress occludes blood flow. The schematic of experimental apparatus is indicated in Fig. 3a, and red DMEM was first used. More than 2.6 mL DMEM passes through the ex vivo blood vessels before hydrogel–GNR gelation. After gelation, the volume sharply decreases to less than 0.2 mL (Fig. 3b). This intriguing phenomenon robustly supports the hypothesis that gelation shrinkage-derived internal stress can squeeze blood vessels and give rise to vascular embolism. Subsequently, we evaluated the ability of vascular embolism to occlude blood supply that is indispensible for favoring tumor progression. Similarly, due to blood supply occlusion, the volume of gathered blood that passes through the ex vivo blood vessels sharply decreases after gelation (Fig. 3c) and the numbers of red

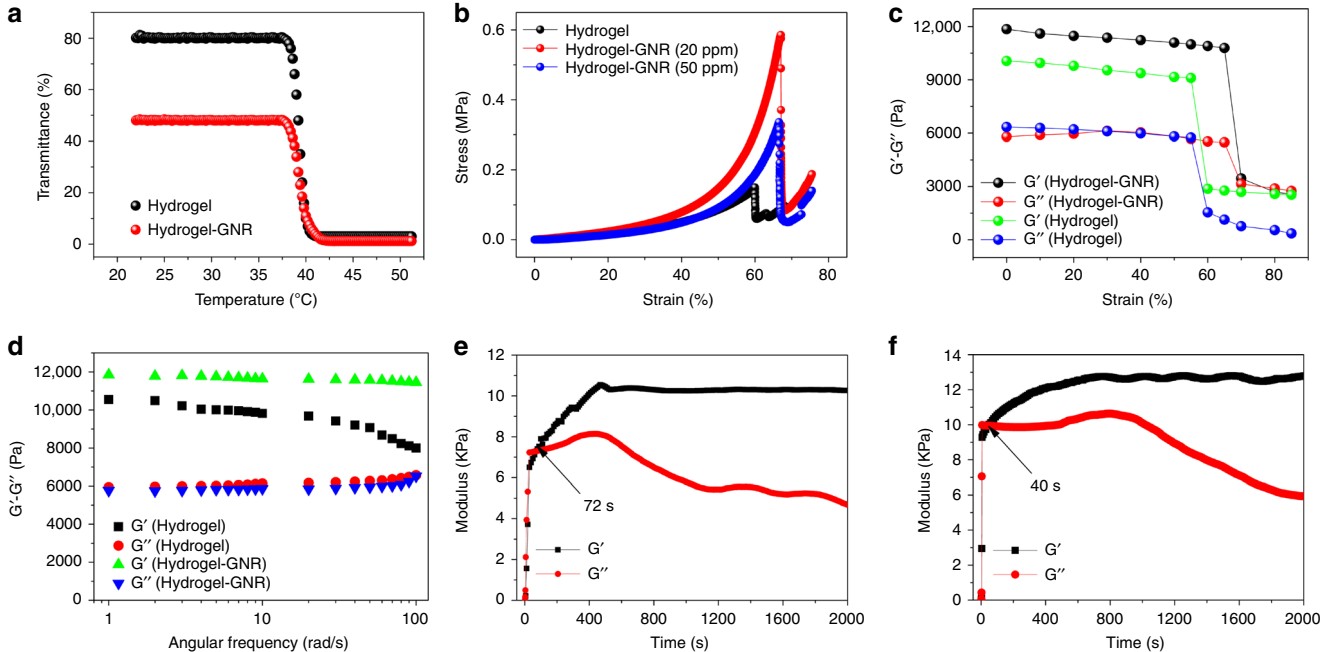

**Fig. 2** Rheological measurements of composite hydrogel and hydrogel–GNR. **a** Temperature-dependent transmittance of hydrogel and hydrogel–GNR, and the temperature corresponding to the abrupt drop point is designated as the LCST. **b** Compressive stress–strain curves of hydrogel, hydrogel–GNR (20 ppm), and hydrogel–GNR (50 ppm), respectively. **c** Strain amplitude sweeps of hydrogel and hydrogel–GNR performed at 39 °C at a constant frequency of 1 rad s$^{-1}$, **d** Dynamic oscillatory frequency sweeps of hydrogel and hydrogel–GNR at a fixed strain of 1%. **e, f** Time-dependent sweep tests of hydrogel (**e**) and hydrogel–GNR (**f**). Notes: hydrogel represents CS/mPEG-Mal/pNIPAAm-*co*-AAc with 20% AAc in pNIPAAm-*co*-AAc.

blood cells, blood platelets, and white blood cells in the collection container is much less than those before gelation (Fig. 3d–f)[36].

In vivo blood flow occlusion of normal blood vessels in abdomen was also monitored, and color Doppler flow imaging (CDFI) technology was used to investigate the in vivo blood vessel squeezing, and the schematic is depicted in Fig. 3g. Rich and rapid blood stream and wide blood vessel are observed in an abdomen artery pre-exposure to laser irradiation (Fig.3h and Supplementary Movie 1). In contrast, once exposure to 808 nm laser, the laser irradiation rapidly triggers the gelation process of hydrogel–GNR via GNR-mediated temperature elevation, simultaneously accompanied by internal stress. Such an extravascular internal stress successfully squeezes and narrows the blood vessel (indicated by arrows), resulting in the successful occlusion of blood supply (Fig. 3h and Supplementary Movie 2).

**Intratumoral vascular narrowing and occlusion test**. To further comprehend the underlying principle of this extravascular gelation shrinkage-derived internal stress for starvation therapy, in vivo intratumoral vascular narrowing and blood & nutrition supply occlusion tests on PANC-1 xenografted tumor implanted on nude mice were explored. Blood supply in blood vessels of tumor and its adjacent tissues is considerably decreased after instant treatment with Laser+hydrogel–GNR (2) (Fig. 4a and Supplementary Movies 3 and 4), suggesting the occurrence of vascular narrowing. Even after 5 days, the occluded blood supply remains unrecovered (Fig. 4a and Supplementary Movie 5), which is beneficial for the persistent starvation therapy.

To figure out whether in vivo vascular narrowing arising from the gelation shrinkage-mediated internal stress for squeezing blood vessels is responsible for the occluded blood supply in tumor, in vivo perfusion studies with FITC-labeled dextrans with varied sizes (i.e., 4, 20, and 70 K) as tracers were first carried out. It is found that far lower fluorescence signal intensity at tumor in

treated group (i.e., Laser+hydrogel–GNR) than that in control group (i.e., Laser+GNR) suggests that dextrans are not allowed to enter the tumor due to vascular occlusion (Fig. 4b, c)[36]. Even though the molecular weight of dextran drops as low as 4 K, these FITC-labeled dextrans in treated group fail to enter tumor. This result sufficiently demonstrates that vascular narrowing in tumors is responsible for the blood flow drop and blood supply occlusion rather than the increased intratumoral pressure.

Moreover, in vivo perfusion studies with clinically used Sonovue$^{TM}$ microbubbles (MBs) as tracers that can be detected under the contrast-enhanced harmonic imaging (CHI) mode was further implemented. Before corresponding treatments in control and treated groups, the 1st injection of Sonovue$^{TM}$ MBs illuminates the tumors in both groups (Fig. 4d, e and Supplementary Movies 6 and 8)[36]. After corresponding treatments that were carried out after 30–90 min post-1st injection of Sonovue$^{TM}$ MBs for guaranteeing collapse of the 1st injected Sonovue$^{TM}$ MBs, the 2nd injection of Sonovue$^{TM}$ MBs illuminates the tumor in control group, but fails in treated group (Fig. 4d, e and Supplementary Movies 7 and 9). This result further demonstrates that the Laser+hydrogel–GNR treatment in treated group results in vascular narrowing due to the extravascular gelation shrinkage-derived internal stress in tumor.

To directly observe the vascular narrowing phenomenon, laser confocal scanning microscopic (LCSM) images of tumor slices stained by CD31 and CD34 immunofluorescence that characterize new and matured blood vessels, respectively, were captured. Laser+hydrogel–GNR treatment in treated group indeed induces the narrowing of both budding and matured blood vessels in tumor compared with the Laser+GNR treatment in control group (Fig. 4f, g). These intriguing results convincingly demonstrate that extravascular gelation shrinkage-derived internal stress can squeeze and narrow blood vessels in tumor and subsequently result in blood supply occlusion.

Noticeably, excellent diffusion is a robust guarantee of realizing this special starvation therapy. It is assured that this injectable

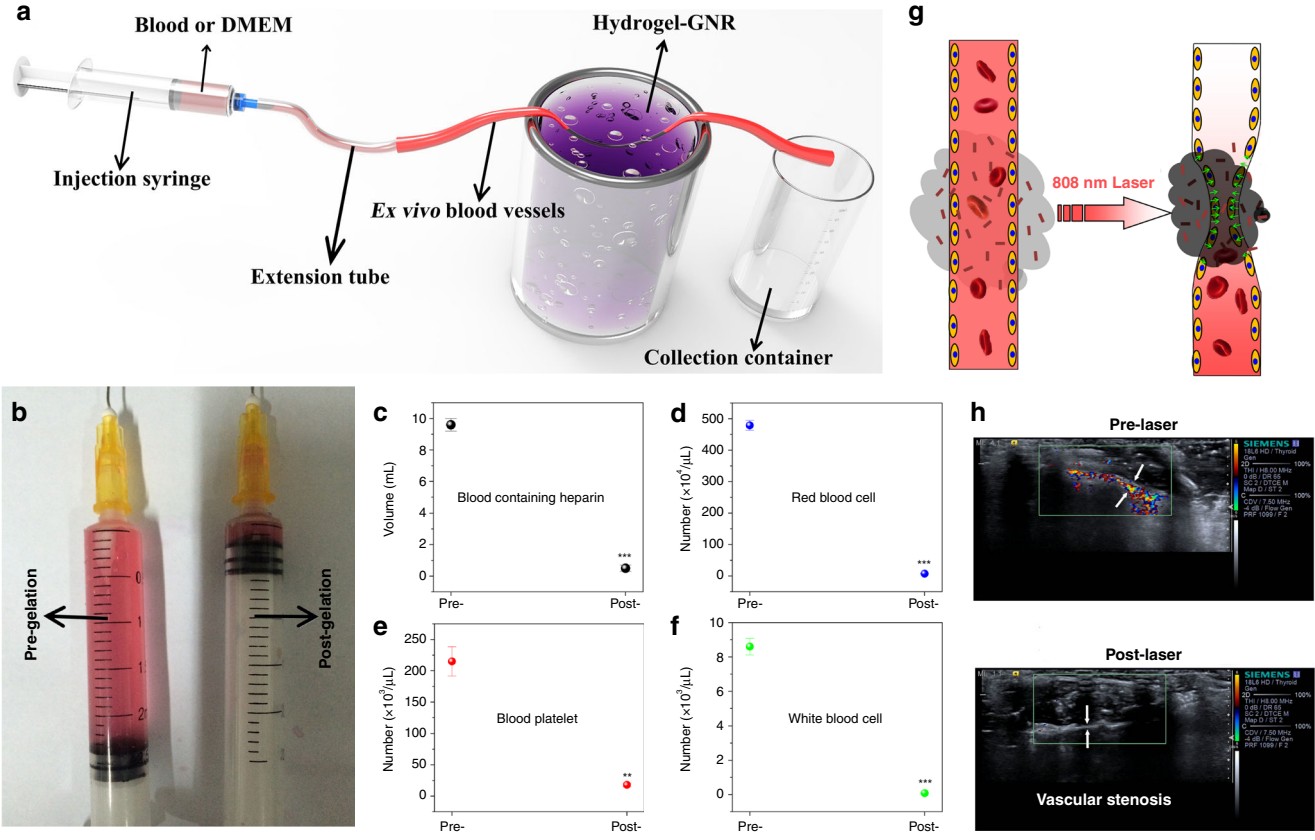

**Fig. 3** Occlusion tests of blood supply on ex vivo and in vivo artery model. **a** Experimental apparatus schematic available for evaluating the occlusion outcome of ex vivo blood vessels with an inner diameter of 1.00 mm based on the extravascular gelation shrinkage-induced internal stress for squeezing blood vessels. **b** Traversed volume of DMEM through the ex vivo blood vessels before (left) and after (right) gelation of hydrogel–GNR. **c, f** Collected blood volume (**c**), red blood cells (**d**), blood platelet (**e**), and white blood cells (**f**) traversed through ex vivo blood vessels using rabbit blood containing heparin instead of above DMEM. Data are expressed as mean value ± SD ($n = 3$). **$P < 0.01$ and ***$P < 0.001$, and the statistical significances were obtained using student's $t$ test in comparison to Pre-gelation. **g** In vivo schematic of extravascular gelation shrinkage-induced internal stress for squeezing blood vessels upon exposure to 808 nm laser. **h** CDFI images of abdomen artery of nude mice around which hydrogel–GNR solution was administrated, and in particular, CDFI images were captured before and after 808 nm laser irradiation, respectively, wherein white arrows indicate the blood vessels of abdomen artery.

composite (i.e., hydrogel–GNR) features excellent diffusion, since 500 nm silica particles were reported to easily pervade the whole tumor[45]. As well, in vivo fluorescence imaging (Fig. 4h) and MR imaging (Supplementary Fig. 10) indicate that the injectable hydrogel–GNR can rapidly diffuse and pervade the whole tumor within 1 min, wherein Cy3-labeled dextran (70 K) and extensively accepted hollow porous $Fe_3O_4$ nanoparticles instead of GNRs (Supplementary Fig. 10a,b) served as the fluorescence and T2-weighted MRI tracers, respectively[46].

**In vivo starvation therapy on PANC-1 xenografted tumor.** Inspired by above results on in vivo vascular narrowing & embolism and blood & nutrition supply occlusion, exploration using this special starvation therapy on PANC-1 tumor was carried out. Analogous to in vitro results, 808 nm laser is also applicable for triggering intratumoral gelation process, since it can elevate the temperature above 39 °C (Supplementary Fig. 11). In particular, a special administering method represented by Laser+hydrogel–GNR (2) wherein hydrogel–GNR were injected into tumor and its periphery, was employed to simultaneously cut off the blood & nutrition supply in tumor and its adjoining normal tissues (Fig. 5a). The conventional injection method (i.e., intratumoral injection) represented by Laser+hydrogel–GNR (1) plays an inhibitory effect against PANC-1 tumor to some extent (Fig. 5b)[36]. As a contrast, the special administering method harvests the largest inhibitory

outcome, and the tumor volume approximately levels off to a plateau (Fig. 5b), meaning no increase. Digital photos also indicate negligible tumor growth when treated with Laser+hydrogel–GNR (2) (Supplementary Fig. 12). The therapeutic difference between two groups can be attributed to that the robust internal stress and compact structure of composite hydrogel–GNR in Laser+hydrogel–GNR (2) group can squeeze and narrow blood vessels, induce embolism and occlude blood & nutrition supply in both tumor and its adjacent normal tissues around tumor periphery, while Laser+hydrogel–GNR (1) can merely work in tumor.

Noticeably, another comparison, i.e., pNIPAAm without AAc, was used and a identical injection manner with Laser+hydrogel–GNR (2) was employed. pNIPAAm can spontaneously give rise to gelation due to its low LCST (32 °C) below the body temperature. pNIPAAm gels accommodate loose structure with macropores (Supplementary Fig. 13a) and share poor mechanical strength, elasticity and rheological property (Supplementary Fig. 13b–d). More significantly, re-illumination after experiencing the 2nd injection of Sonovue™ MBs (Supplementary Fig. 14) and evident fluorescence signal (Supplementary Fig. 15) at tumor are observed in treated group, akin to control group. This phenomenon suggests that pNIPAAm is disabled to resist blood pressures, narrow blood vessels and occlude blood & nutrition supply in tumor consequently failing to delay tumor growth (Fig. 5b). However, it indirectly suggests that the compact structure and robust internal stress in Laser+hydrogel–GNR (2) or Laser+

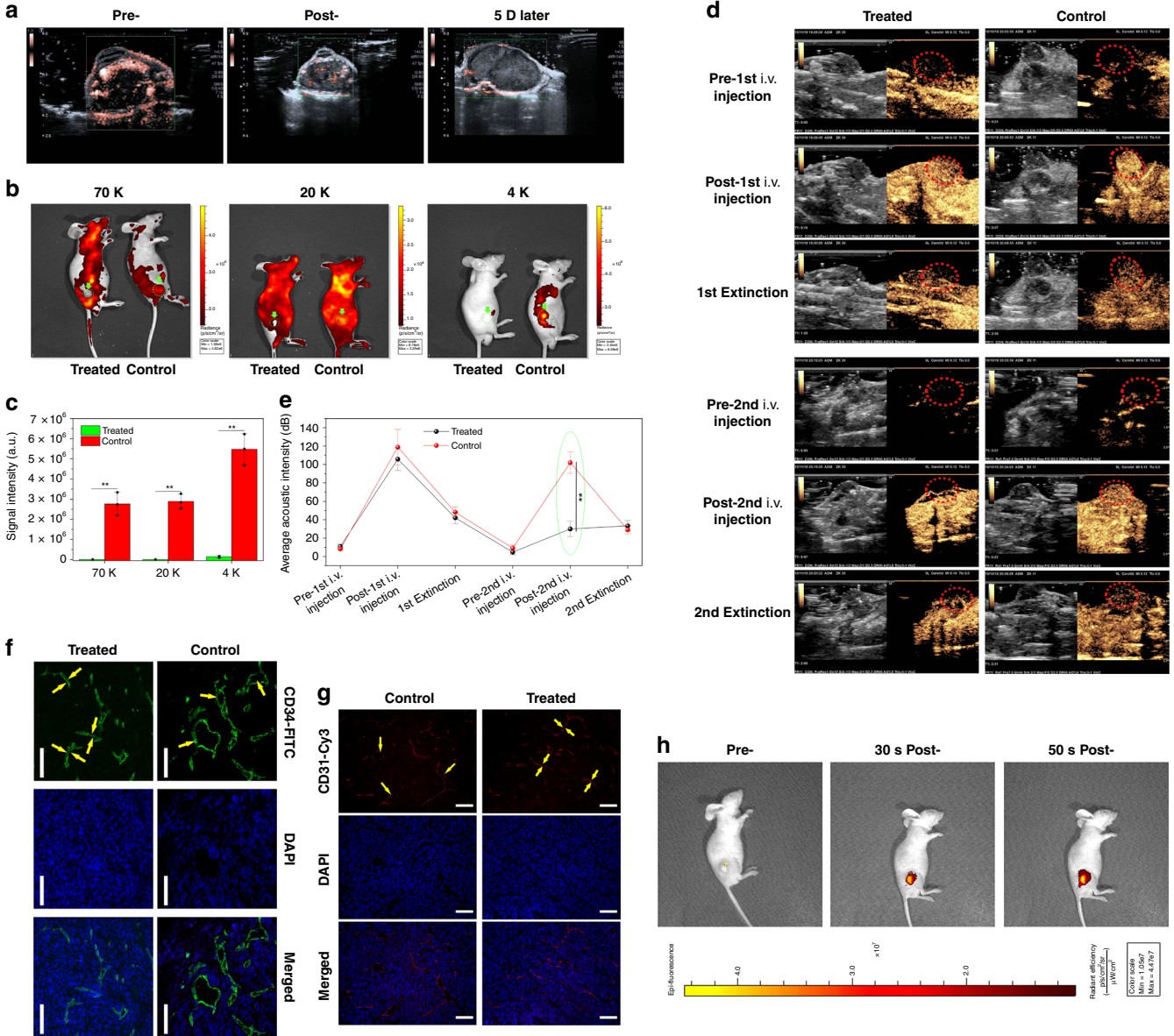

**Fig. 4** In vivo evaluations on intratumoral vascular narrowing on PANC-1 xenografted pancreatic tumors. **a** CDFI images of PANC-1 xenografted pancreatic tumors implanted on nude mice that were captured at three time points, i.e., pre-, post- and 5 day post-treatment with Laser+hydrogel–GNR (2). **b**, **c** In vivo animal fluorescence images (**b**) and quantitative signal intensities (**c**) of PANC-1 xenografted tumor after corresponding treatments in control (i.e., Laser+GNR) and treated (i.e., Laser+hydrogel–GNR (2)) groups and subsequent injections of different FITC-labeled dextrans with varied sizes (i.e., 70, 20, and 4 K), wherein green arrows and dotted ellipses indicate the tumor. Data are expressed as mean ± SD ($n = 3$). **d**, **e** Time-dependent ultrasonic images (**d**) and average acoustic intensities (**e**) of PANC-1 xenografted tumor before and after corresponding treatments in control and treated groups, wherein the 1st i.v. injection of Sonovue™ microbubbles were carried out, and after 30–90 min, the corresponding treatments in two groups, i.e., control (Laser+GNR) and treated (Laser+hydrogel–GNR (2)), were implemented, followed by the 2nd i.v. injection of Sonovue™ microbubbles; and the ultrasonic images were captured at three time points, i.e., pre-, post-, and 40–120 s post- each i.v. injection of Sonovue™ microbubbles, and red dotted ellipses indicate the tumor. Data are expressed as mean ± SD ($n = 3$). **f**, **g** Matured (**f**) and budding (**g**) blood vessels of PANC-1 xenografted tumors after corresponding treatments in control and treated groups, and they were stained by FITC-labeled CD34 immunofluorescence (**f**) and Cy3-labeled CD31 immunofluorescence (**g**), respectively, scale bar: 200 μm. Yellow arrows indicate the intratumoral blood vessels. (**h**) In vivo animal fluorescence imaging of nude mice-bearing PANC-1 xenografted tumor at three time points, i.e., pre-, 30 s post- and 50 s post-i.t. injection of Cy3-labeled dextrans-grafted hydrogels, wherein Cy3-labeled dextran (size: 70 K) instead of GNRs was used. Statistical significances were obtained using unpaired student's $t$ test, **$P <$ 0.01. Note: Laser+hydrogel–GNR (2) represent the co-injection of hydrogel–GNR in tumor and its periphery and subsequent 808 nm laser irradiation.

hydrogel–GNR (1) are responsible for the successful starvation therapy. After treatment with Laser+hydrogel–GNR (2), the survival time of nude mice is significantly prolonged (Fig. 5c) and no evident weight variation is observed (Supplementary Fig. 16)[36].

To better understand the underlying principle of this extravascular gelation shrinkage-derived internal stress for

starvation therapy in Lase+hydrogel–GNR (2), several pathological examinations including H&E, PCNA and CD31 immuno-histochemical assays and CD34&TUNEL immunofluorescence co-staining assay were implemented. Some evident characteristics associated with apoptosis, e.g., chromatin condensation, nucleus disintegration and cell lysis, are observed (Fig. 5d) and

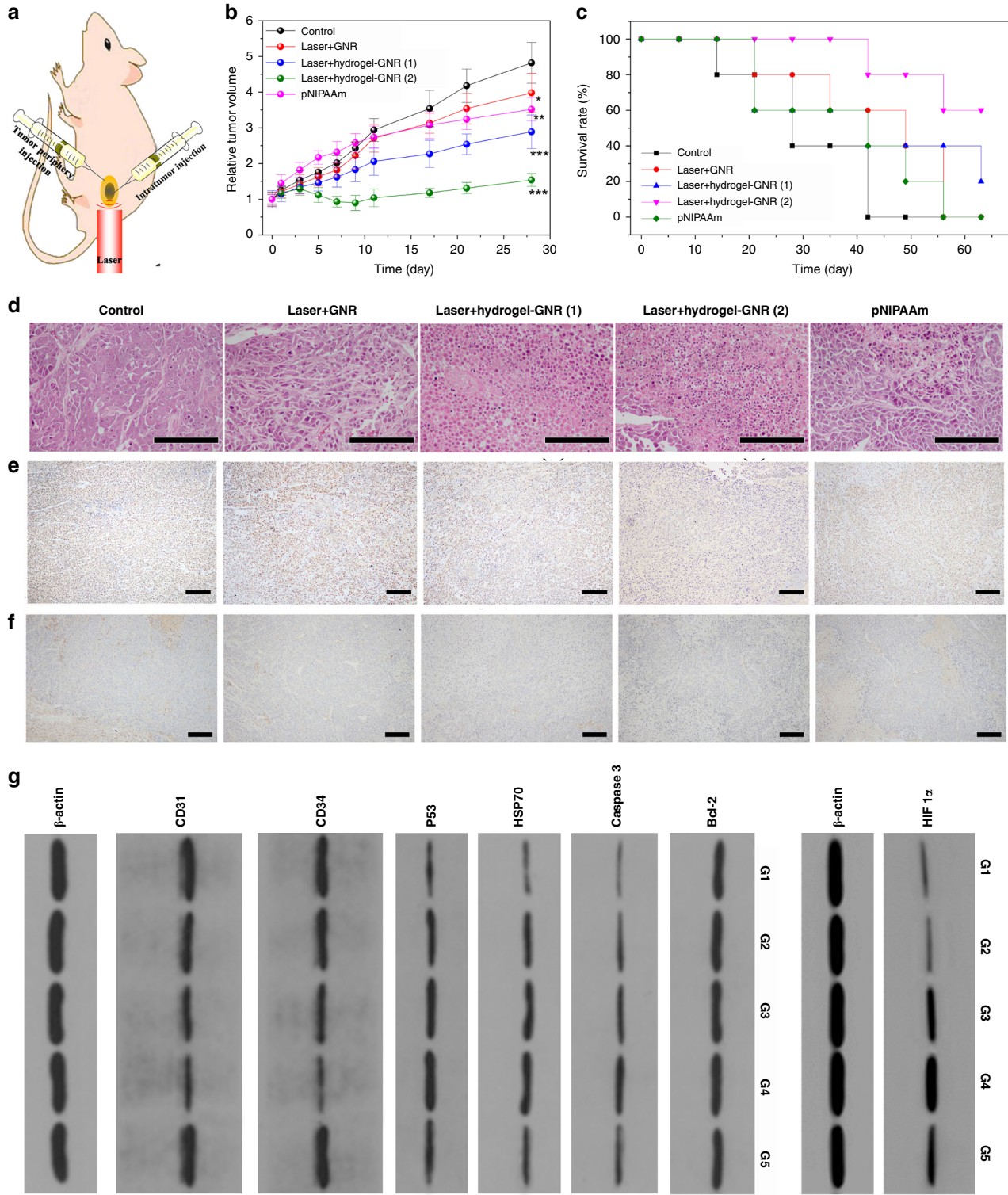

Laser+hydrogel–GNR (2) treatment in G4 induces the largest apoptosis (Supplementary Fig. 17a). More significantly, this treatment also acquires the largest inhibitory effects against PANC-1 cell proliferation and CD31-labeled angiogenesis via cutting off nutrition supply due to vascular angiostenosis and occlusion (Fig. 5e, f and Supplementary Fig. 17b,c)[36]. Moreover, Laser+hydrogel–GNR (2) treatment in G4 brings about the largest apoptotic cells (62%) including tumor cells (82%) and endothelial cells (18%), and simultaneously results in the lowest CD34-labeled vascular density (Fig. 6a–c)[36]. Thus, this special

starvation therapy harvests the most excellent treatment outcome through inducing apoptosis, inhibiting cell proliferation and CD31-labeled angiogenesis and reducing CD34-labeled vascular density.

Furthermore, western blot analysis was carried out to explore the apoptotic mechanism. The persistent blood supply occlusion due to vascular narrowing in this special starvation therapy (i.e., G4) is found to significantly up-regulate hypoxia inducible factor 1α (HIF 1α) and simultaneously up-regulates the pro-apoptotic protein (i.e., P53) (Fig. 5g and Supplementary Fig. 18)[36]. This

**Fig. 5** In vivo evaluation of this starvation therapy in inhibiting PANC-1 pancreatic tumors on nude mice. **a** Schematic of in vivo experimental process, and hydrogel–GNR was simultaneously injected into both tumor and its periphery where 808 nm laser irradiation was carried out. **b, c** Time-dependent tumor volume variation (**b**) and survival rate (**c**) of PANC-1 tumor-bearing nude mice experiencing different treatments, i.e., control, Laser+GNR, Laser+hydrogel–GNR (1), Laser+hydrogel–GNR (2) and pNIPAAm, wherein (1) and (2) represent tumor injection alone and co-injection of tumor and its periphery, respectively, and pNIPAAm was also injected into both tumor and its periphery. Data are expressed as mean ± SD (*n* = 6); \*P < 0.05, \*\*P < 0.01 and \*\*\*P < 0.001, and the statistical significances were calculated via unpaired Student's *t* test in comparison to control. **d–f** Microscopic images of PANC-1 tumor slices stained by H&E immunohistochemical staining (**d**), PCNA immunohistochemical staining (**e**) and CD 31 immunohistochemical staining (**f**) that were harvested after sacrificing of PANC-1-bearing nude mice treated with aforementioned different treatments (G1-G5) on the 10th day post-treatment, scale bar: 200 μm. Notes: H&E, PCNA and CD31 immunohistochemical assays were employed to evaluate cell structure, cell proliferation, and neovascularization, respectively. **g** Western blot analysis for analyzing different proteins including Bcl-2, Caspase 3, HSP70, P53, CD34 and CD31 and HIF 1α in PANC-1 tumors receiving aforementioned different treatments (G1–G5) on the 10th day post-treatment. Note: G1–G5 represent control, Laser+GNR, Laser+hydrogel–GNR (1), Laser+hydrogel–GNR (2) and pNIPAAm, respectively.

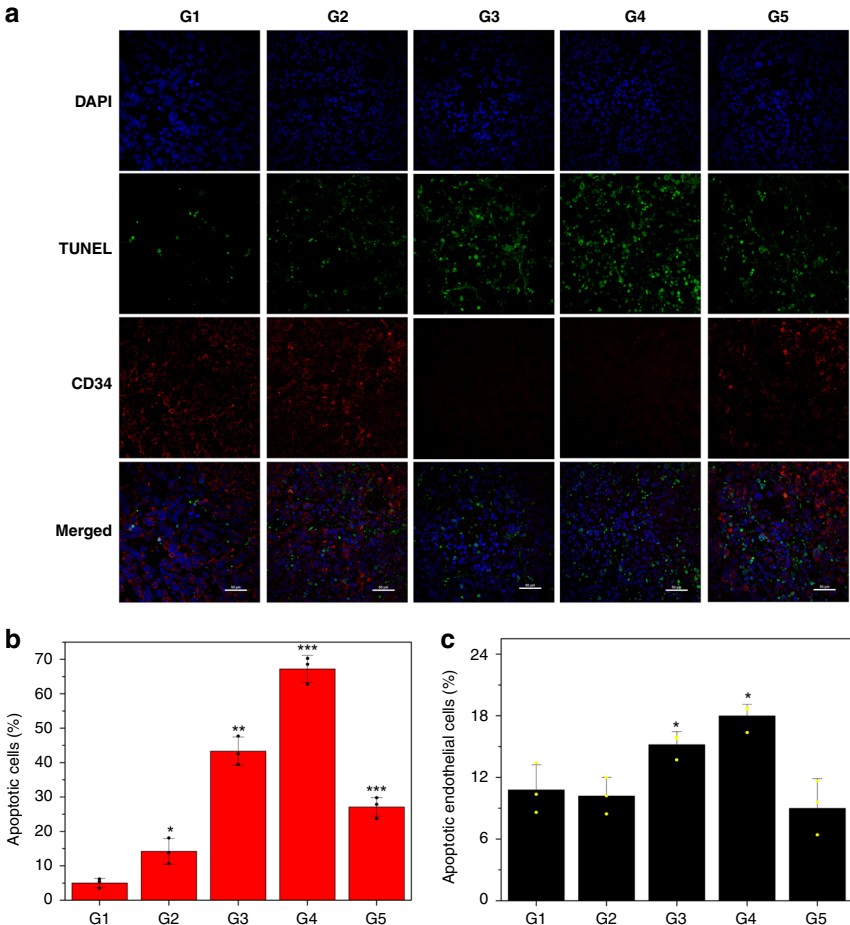

**Fig. 6** In vivo principle exploration of anti-tumor using this special treatment method. **a** LCSM images of PANC-1 tumor slices co-stained by TUNEL immunofluorescence & CD34 immunofluorescence that were harvested after sacrificing of PANC-1-bearing nude mice treated with aforementioned different treatments (G1-G5) on the 10th day post-treatment, scale bar: 50 μm, and TUNEL & CD34 immunofluorescence co-staining was employed to determine cell apoptosis and vascular density, respectively, wherein FITC (green) and TRITC (red) were used to label TUNEL and CD34, respectively, and nuclei (blue) was stained by DAPI. **b, c** Semi-quantitative data of apoptotic cells (**b**) and apoptotic endothelial cells in all apoptotic cells (**c**) after different treatments with G1–G5 via calculating the ratios of green-labeled cells in whole blue-labeled cells and red and green co-labeled cells in all green-labeled cells according to the TUNEL immunofluorescence staining, respectively. Data are expressed as mean ± SD (*n* = 3). \*P < 0.05, \*\*P < 0.01 and \*\*\*P < 0.001, and the statistical significances were obtained using student's *t* test in comparison to G1. Note: G1-G5 represent control, Laser+GNR, Laser+hydrogel–GNR (1), Laser+hydrogel-GNR (2) and pNIPAAm, respectively.

result is completely consistent with previous report focusing on intravascular aggregate-mediated blood & nutrition occlusion for starvation therapy[10]. Noticeably, no expression variation of Bcl-2 pro-apoptotic protein indicates HIF 1α elevation can trigger apoptosis through the pathway, i.e., the stabilization of P53[47,48]. Intriguingly, the largest down-regulations of CD34 and CD31 proteins in G4 (i.e., Laser+hydrogel–GNR) further demonstrate

that this special treatment can induce endothelial cell apoptosis and exert a robust influence on anti-angiogenesis via cutting off blood & nutrition supply. This result is completely consistent with previous TUNEL & CD34 immunofluorescence co-staining and PCNA and CD31 immunofluorescence staining. Therefore, it is not difficult to understand why this starvation therapy harvests the highest apoptotic cells and the strongest inhibitory effects

against cell proliferation, angiogenesis and tumor growth (Figs. 5 and 6), since high expressions of both HIF 1α and P53 are essential for the hypoxia-induced cell death[49–51]. In particular, the ranking orders of anti-proliferation, anti-neovascularization, cell apoptosis and expressions of pro-apoptotic proteins and HIF 1α are consistent with that of aforementioned therapeutic outcome in all groups.

Taken all above experimental results together, such a special starvation therapy based on extravascular gelation shrinkage-derived internal stress can squeeze and narrow blood vessels (Fig. 4f, g), occlude blood & nutrition supply (Fig. 4a–e), induce hypoxia (Fig. 5g), starve tumor cells and endothelial cells to death (Figs. 5d–g and 6), reduce vascular density (Fig. 6) and inhibit PANC-1 proliferation and angiogenesis (Fig. 5e, f), which eventually suppresses tumor growth (Fig. 5b). The whole procedure regarding this underlying principle is indicated in Supplementary Fig. 19.

**In vivo starvation therapy on 4T1 xenografted tumor.** To test the generality of this special starvation therapy, another model, i.e., 4T1 breast cancer implanted on immune-competent BALB/c female mice, was used. Contributed by vascular narrowing and blood & nutrition supply occlusion, the treatment with Laser +hydrogel–GNR (1) or (2) exerts prominently inhibitory effects on tumor growth. In particular, Laser+hydrogel–GNR (2) wherein tumor and its adjoining tissues are treated performs the best in suppressing tumor growth (Fig. 7a, b)[36]. In contrast, pNIPAAm fails to delay tumor growth due to the poor internal stress and loose structure. These results sufficiently suggest that this special starvation therapy can be extended to other tumor models. During experimental period, no evident weight variation of mice is observed (Fig. 7c)[36].

As well, in vivo intratumoral vascular narrowing and blood & nutrition supply occlusion tests were explored on 4T1 xeno-grafted tumor. In vivo perfusion studies with FITC-labeled dextrans and Sonovue$^{TM}$ MBs as tracers of fluorescence imaging and ultrasound contrast imaging, respectively, were first carried out. Akin to PANC-1 model on nude mice, after treatment with Laser+hydrogel–GNR, dextrans with varied sizes (i.e., 4, 20, 70 K) and the 2nd injected Sonovue$^{TM}$ microbubbles fail to enter tumor in comparison to control group (i.e., Laser+GNR) (Fig. 7d–g and Supplementary Movies 10–13)[36]. These results confirm again that the constricted vessel narrowing arising from the gelation shrinkage internal stress-mediated squeezing against blood vessels in 4T1 tumor and adjacent tissues is responsible for the occluded blood & nutrition supply and suppressed tumor growth.

More significantly, this starvation therapy exerts robust influences on inflammatory microenvironment in tumor. In detail, Laser+hydrogel–GNR (2) in G4 suppresses the secretions of IL1, IL6, IL8, IL10, IL12, CCL22, CCL17, and CCL5 that can promote progression and metastasis of tumor (Fig. 7h) and simultaneously augments the secretions of IL-5, IL-18, IL21, IFN-γ, and TNF-α associated with anti-tumor activity (Supplementary Fig. 20)[36]. In addition, vascular embolism, reduced vascular density and occluded migration passages in this starvation therapy also hamper the infiltrations of macrophages (CD11b +F480+), dendritic cells (DC, Cd11b-CD11c+) and leukocytes cells (CD45+) in tumor (Fig. 7i and Supplementary Fig. 21)[36]. Notably, this starvation therapy fails to significantly vary the sub-phenotypes of CD45+ cells (e.g., natural killing cells, T cells including Th and Tc) (Supplementary Fig. 22)[36].

**In vitro inhibitory tumor metastasis test.** Tumor metastasis is another important character of malignancy[1,52,53]. Great progress

has been made in inhibiting tumor metastasis, e.g., nanoparticle endocytosis-mediated anti-metastasis, GNRs photothermal-mediated anti-metastasis[54,55], and targeted binding-mediated anti-metastasis[56,57]. Herein, the well-proven reduced porosity and pore diameter by internal stress-induced shrinkage in hydrogel–GNR are expected to occlude migration pathway of tumor cells and hamper tumor cell invasiveness and migration into blood vessels. To demonstrate it, in vitro anti-metastasis experiment was carried out and the apparatus schematic is shown in Fig. 8a. Primary tumors were seeded in culturing zone designated as S0, and S1–S4 are designated as the distant transferring zones. The separation wall between S0 and either one in S1–S4 is made up of hydrogel–GNR. It is clearly found that the number of PANC-1 cells in S0 increases as a function of incubation time, but there are still no evident PANC-1 cells in S1–S4 (Fig. 8b). Quantitatively, as the incubation time proceeds, the cell density continuously rises from $3.2 \times 10^5$ to $12 \times 10^5$ per mL after 70 h, while a cell density less than 20 per mL is obtained (Fig. 8c, d)[36]. These results suggest that hydrogel–GNR after gelation indeed impeded in vitro tumor cell migration and invasion from S0 to S1–S4.

**In vivo inhibitory tumor metastasis test.** In particular, vascular stenosis, blood occlusion, and reduced vascular density resulting from the extravascular gelation shrinkage of hydrogel–GNR in such an extravascular gelation shrinkage-mediated starvation therapy can blockade the migration passage of tumor cells, which is expected to inhibit in vivo metastasis of late-stage tumor through gel matrix (Zone A in Fig. 8e). In contrast, some late-stage tumors or some physical therapies (e.g., photothermal and ultrasound-based therapy) that can destroy blood vessels in tumor facilitate tumor metastasis (Zone B)[11,41]. To demonstrate it, a transgenic mouse model that can generate mammary adenocarcinomas after induction by MMTV-PyVmT oncogene was used as the ideal model for evaluating anti-metastasis ability of this special treatment (Supplementary Fig. 23), since secondary metastatic tumors are easily accessible in the lung[27–29]. The underlying principles of anti-metastasis using this special starvation therapy are displayed in Fig. 8f wherein the Laser +hydrogel–GNR treatment in treated group is expected to cuff off the metastasis pathway via reducing vascular density and occluding blood vessels. In Fig. 8g, no secondary metastatic tumor is observed in lung in treated group, while lots of secondary metastatic tumors in lung emerge in control group. Quantitatively, all mice-bearing mammary adenocarcinomas generate lung metastasis with $9.28 \pm 1.60$ (mean ± standard deviation) lung nodules in control group. In contrast, only 3 mice ($n = 7$) generate lung metastasis with $0.57 \pm 0.79$ lung nodules in treated group (Fig. 8h)[36]. The effectively suppressed lung metastasis can be attributed to the reduced vascular density and occluded migration passage accompanied in this special starvation therapy. Noticeably, this starvation therapy also delays the growth of primary tumor implanted in the orthotopic breast of the transgenic mice (Supplementary Fig. 23).

As well, to further evaluate the in vivo anti-metastasis using this starvation therapy, ultrasound irradiation uniting microbubbles (MBs) that is designated as US(MBs) serves as a means capable of triggering PANC-1 metastasis according to a previous report[11]. It is clearly found that the treatment with US(MBs) alone results in evident distant metastasis (Supplementary Fig. 24b) and the metastasis rate reach approx. 85% (Supplementary Fig. 24d). In contrast, no evidently distant metastasis is observed in the experimental group (i.e., Laser+hydrogel–GNR+US(MBs)) (Supplementary Fig. 24a,c). This phenomenon sufficiently demonstrates that the dual functions of hydrogel–GNR, i.e., inhibitory migration

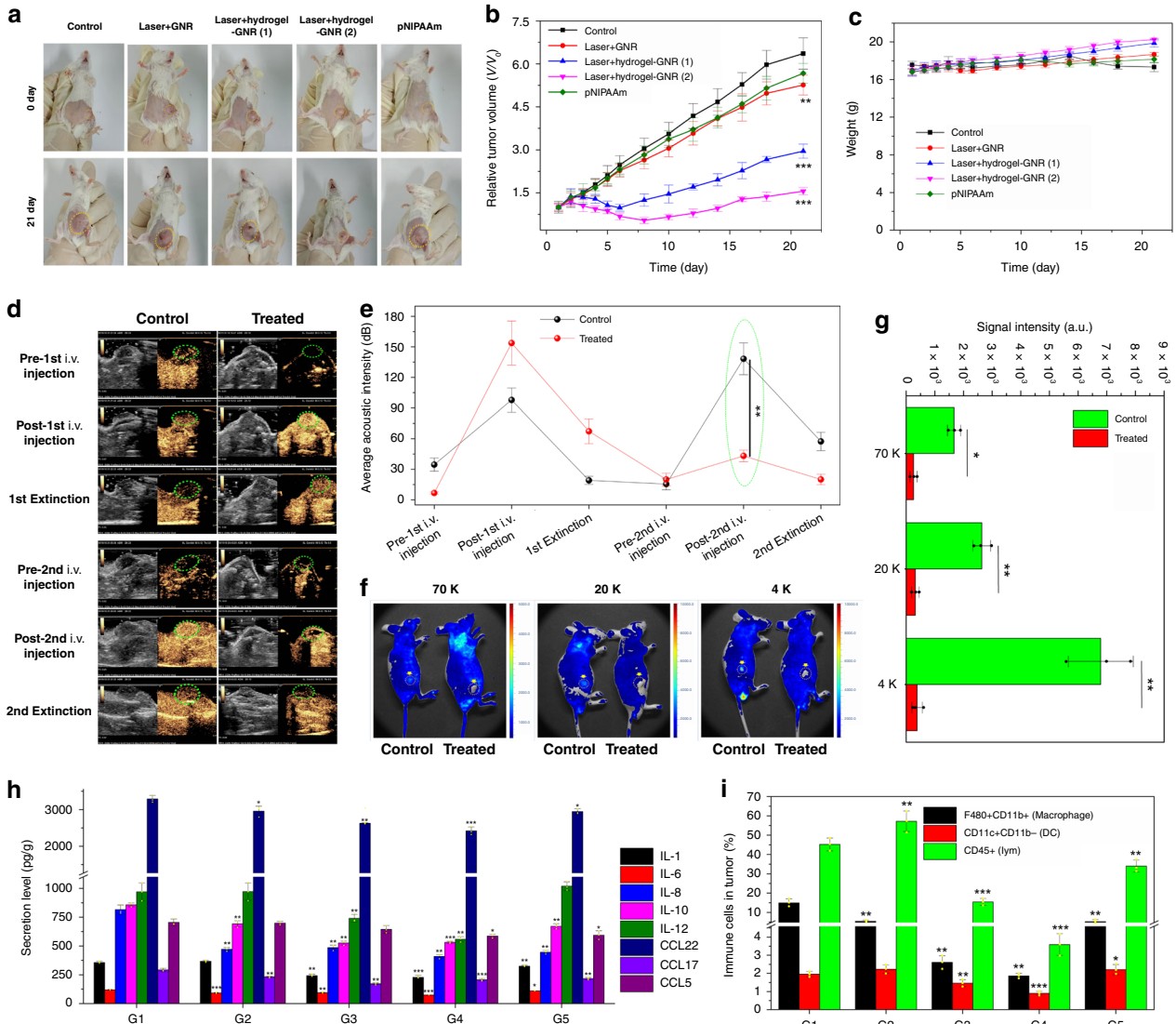

**Fig. 7** In vivo starvation therapy against 4T1 breast tumors subcutaneously implanted on BALB/c mice. **a–c** Digital photos of 4T1 xenografted breast tumors subcutaneously implanted on immune-competent BALB/c mice experiencing different treatments at 0 day and 28 day, i.e., control, Laser+GNR, Laser +hydrogel–GNR (1), Laser+hydrogel–GNR (2) and pNIPAAm, and yellow dotted ellipse indicates the tumor. **b**, **c** Time-dependent tumor volume (**b**) and weight (**c**) variation profiles of 4T1 xenografted breast tumors implanted on BALB/c mice after experiencing aforementioned different treatments. Error bars are based on SD ($n = 6$); **$P < 0.01$ and ***$P < 0.001$, which were calculated via unpaired Student's $t$ test in comparison to the control group. **d**, **e** Time-dependent ultrasonic images (**d**) and acoustic intensities (**e**) of 4T1 tumor before and after corresponding treatments in control (i.e., Laser +GNR) and treated (i.e., Laser+hydrogel–GNR (2)) groups, wherein Sonvue™ microbubbles that can illuminate tumor were used to evaluate the intratumoral occlusion, and green dotted ellipse indicates the tumor, and data are expressed as mean ± SD ($n = 3$). **f**, **g** In vivo animal fluorescence images (**f**) and signal intensities (**g**) of 4T1 tumor after corresponding treatments in control (i.e., Laser+GNR) and treated (i.e., Laser+hydrogel–GNR (2)) groups and subsequent injection of different FITC-labeled dextrans with varied sizes (i.e., 70, 20, and 4 K), wherein yellow arrows and dotted circles indicate the tumor, and data are expressed as mean ± SD ($n = 3$). *$P < 0.05$ and **$P < 0.01$, which were obtained using student's $t$ test. **h** Secretion levels of different inflammationatory cytokines obtained by ELISA analysis in 4T1 tumors after aforementioned different treatments (G1–G5) at day 10, and data are expressed as mean ± SD ($n = 3$). **i** Quantitative variations of tumor infiltrating including macrophages (CD11b+F480+), dendritic cells (Cd11b-CD11c+) and leukocytes cells (CD45+) in 4T1 tumors after aforementioned different treatments (G1–G5) at day 10, and data are expressed as mean ± SD ($n = 3$). *$P < 0.05$, **$P < 0.01$ and ***$P < 0.001$, which were obtained using student's $t$ test in comparison to G1. Note: G1–G5 represent control, Lase+GNR, Laser+hydrogel–GNR (1), Laser+hydrogel-GNR (2) and pNIPAAm, respectively.

in gel matrix and vascular occlusion, cooperatively contribute to the significantly improved anti-metastasis.

**In vivo inhibitory tumor recurrence**. Tumor recurrence after some treatments (e.g., surgery or heat ablation) is another important threat. With cutting off blood & nutrition supply,

vascular occlusion stemming from this gelation-derived internal stress is expected to inhibit tumor recurrence after photothermal ablation. To verify it, another special injection approach was employed in the experimental group. In detail, GNRs alone and hydrogel–GNR were accordingly injected into PANC-1 tumor and its surrounding periphery, respectively. Afterwards, 808 nm laser irradiations with two different

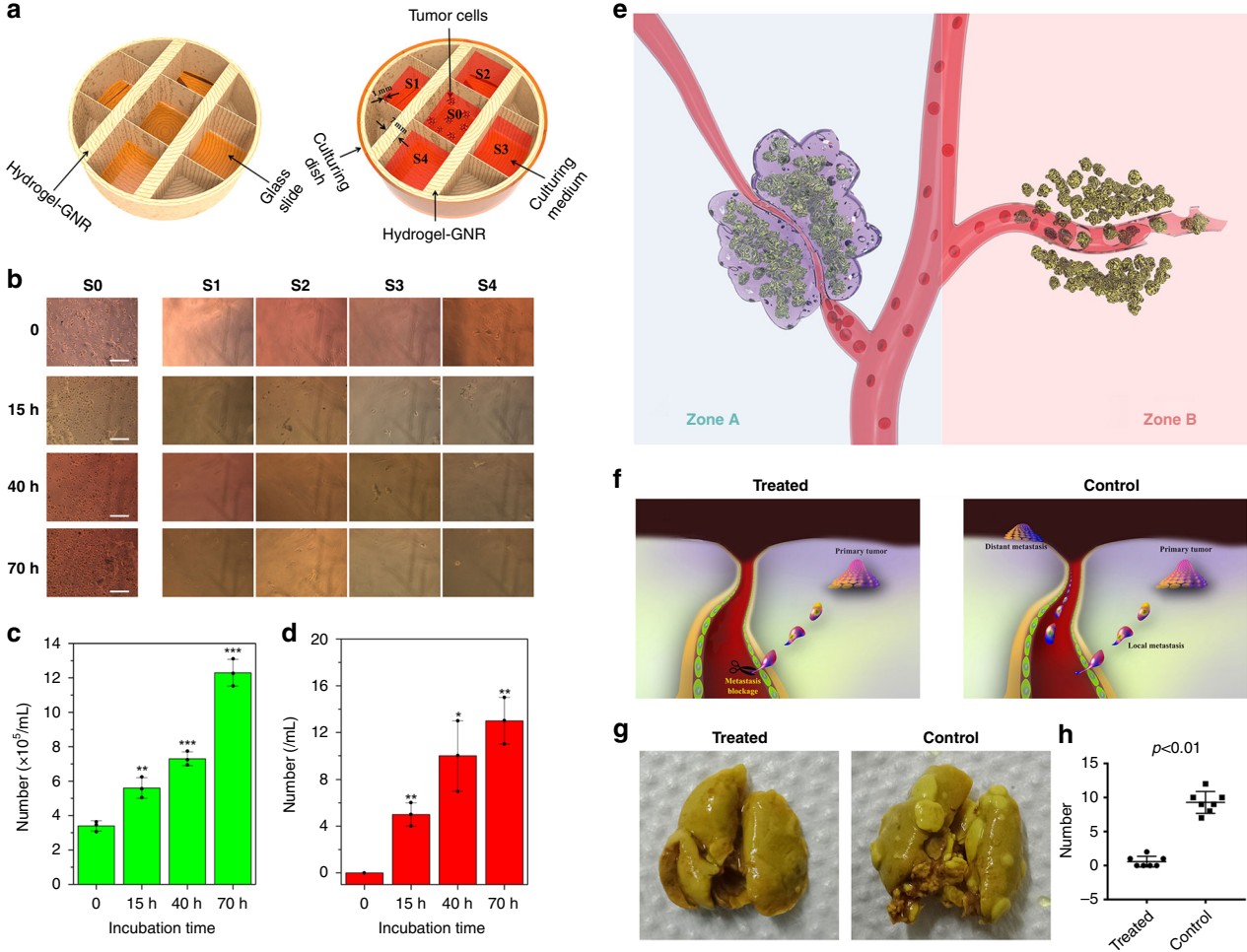

**Fig. 8** Evaluations on inhibited tumor metastasis using this starvation therapy. **a** In vitro anti-metastasis apparatus schematic that is used to evaluate the ability of hydrogel–GNR to occluding the migration passage of cell invasion, wherein S0 is the primary site of PANC-1 tumor cells, and S1–S4 represent four objective metastasis sites and are separated from S0 with hydrogel–GNR with varied thicknesses. **b** Optical microscopic images of S0–S4 after different incubation periods (scale bar: 200 μm). **c, d** The number of time-dependent cells in S0 (**c**) and S1–S4 (**d**). *$P < 0.05$, **$P < 0.01$, and ***$P < 0.001$, which were obtained using student's $t$ test in comparison to 0 h. Data are expressed as mean ± SD ($n = 3$). **e** Principle schematic of in vivo tumor anti-metastasis using such an extravascular gelation shrinkage-induced internal stress to squeeze blood vessels, reduce vascular density and occlude the migration passage of tumor cells; notes: Zone A represents the treated zone with Laser+hydrogel–GNR (2) and Zone B represents the control group without gel occlusion. **f** The underlying anti-metastasis mechanism of this special starvation therapy on the orthotopic mammary adenocarcinomas-bearing transgenic mouse model. **g** Digital photos of lungs acquired from transgenic mice-bearing orthotopic mammary adenocarcinomas in treated and control groups to assess lung metastasis at the end of experiment (day 28). **h** The statistical number of distant tumors in lungs in both control (G1) and treated groups (G4). Values are mean ± SD ($n = 7$) and statistical significances were calculated via unpaired Student's $t$ test. Notes: mice in treated group received Laser+hydrogel–GNR (2) treatment which mean co-injection of hydrogel–GNR in tumor and its periphery and subsequent 808 nm laser irradiation on the 1st day.

laser parameters were instantly carried out for tumor ablation and gelation occurrence (Fig. 9a), respectively. In control group, only GNRs were employed in tumor, followed by 808 nm laser irradiation with larger power density for thermal ablation.

In CDFI images (Fig. 9b), blood supply in the surrounding zone of PANC-1 tumor is significantly decreased, which will cut off nutrition supply and inhibit tumor recurrence. As expected, tumor re-growth occurs to PANC-1 tumors in control group due to the presence of residual tumor cells, while no recurrence is observed in the experimental group (Fig. 9c). Statistically, the recurrence rate drastically drops from 80% in control group to 10% in experimental group (Fig. 9d). This result definitely suggests that the occluded blood supply by this special starvation therapy considerably suppresses tumor recurrence after photo-thermal ablation.

**Biocompatibility and biodegradation evaluations.** Systematic experimental evaluations on the biosafety of hydrogel–GNR are necessary, because excellent biocompatibility and biodegradation are the prerequisites of clinical translation. Even though the residual liquid after gelation reaches 50%, the cell viability attains above 80% (Supplementary Fig. 25)[36]. In in vivo experiments, various liver and kidney function indices and blood biochemical indices indicate no abnormal variation between control group and experimental groups (Supplementary Figs 26, 27)[36]. More-over, H&E images of primary organs in the experimental group (i.e., hydrogel–GNR) also show no evident cell apoptosis or necrosis in comparison to control group (Supplementary Fig. 28). All in vitro and in vivo experiments sufficiently validate the excellent biocompatibility of hydrogel–GNR.

Biodegradability and metabolism of implants are another two concerns before clinical translation. Despite featuring extensively

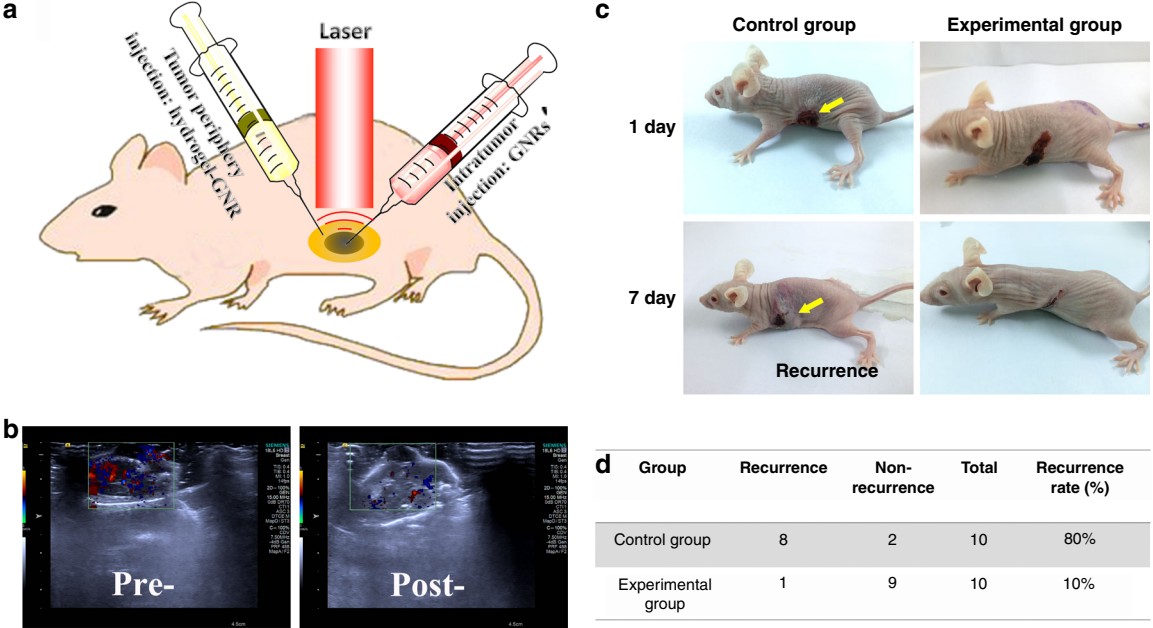

**Fig. 9** Explorations on suppressed tumor recurrence using this starvation therapy. **a** Experiment schematic of tumor recurrence inhibition on PANC-1 tumor-bearing nude mice that received the special treatment wherein hydrogel–GNR was first injected into the surrounding periphery of PANC-1 tumor and laser (1) irradiation was carried out for triggering gelation process, followed by GNRs injection into the PANC-1 tumor and laser (2) irradiation for ablating PANC-1 tumor. **b** CDFI images of PANC-1 tumor and its periphery before and after the special treatment in (**a**). **c** Digital photos of nude mice-bearing PANC-1 solid tumors that experiencing two different treatments, and one is only GNRs injection in tumor uniting laser (2) irradiation for tumor ablation (designated as control group), and another is the special treatment in a (designated as experimental group), wherein yellow arrows indicate the tumor recurrence after ablation. **d** Quantitative statistics of tumor recurrence of PANC-1 tumor implanted on nude mice ($n = 10$) in c.

accepted biosafety and degradability of pNIPAAM-based hydrogels[58–60], evaluating the biodegradability and metabolism of such a composite hydrogel–GNR is necessary. In vitro biodegradation result shows that above 30% hydrogel–GNR gels degrade after 30 days (Supplementary Fig. 29)[36]. In in vivo level, the classic subcutaneous embedding method was used[61]. Intriguingly, a high biodegradation rate of hydrogel–GNR is observed after 60 days post-embedding and the embedded solid hydrogel–GNR approximately vanishes (Supplementary Fig. 30)[36]. As for metabolism, the fate of GNRs that label this composite hydrogel is accessible to indirectly reflect the fates of pNIPAAm-based hydrogels, since the robust covalent bonding between GNRs with hydrogel skeleton makes them fail to shed from hydrogel skeleton during degradation. Over 90% excretions of GNRs through urine and faeces indicate the metabolism pathway of such a composite hydrogel–GNR after degradation (Supplementary Fig. 31)[36].

## Discussion
In anti-tumor experiments against PANC-1 and 4T1 tumors, regrowth occurred after 10 days post-treatment. The failure of complete tumor eradication was attributed to the decomposition and collapse of gel framework based on the fact that the degradation-arising internal stress recession after long-term incubation (Supplementary Figs 30 and 31) brought about gradual recovery of blood & nutrition supply. To address it, future work can follow two pathways. One pathway is designing robust gel framework that can withstand erosion and keep long-term biostability and robust internal stress with compact structure, which will be beneficial for continuously squeezing and narrowing blood vessels and occluding blood & nutrition supply. Another is loading anti-drugs or other therapeutic agents for secondary therapy against tumor via extravascular gelation shrinkage-mediated drug release and degradation-mediated drug release[20,62–64].

As for inhibited recurrence and metastasis, residual tumor cells after surgery or other therapeutic methods probably result in tumor recurrence or metastasis. This starvation strategy can serve as an auxiliary method to address the issues via reducing vascular density, cutting off blood & nutrition supply, occluding the migration passages of tumor cells and starving the residual tumor cells to death before the failure of vascular occlusion (Figs. 8 and 9). Thus, future studies will also focus on designing robust and compact hydrogels with appropriate degradability rate that guarantees continuous work.

Although a preliminary evaluation using this treatment method to inhibit metastasis of late tumors on transgenic mice-bearing breast carcinoma also realizes a suppressed growth for late-stage tumor (Supplementary Fig. 23), systematic evaluations on late-stage tumor treatment are also necessary in future studies to explore the generality of this special starvation therapy. As well, other heat-triggered means (e.g., alternating magnetic field) are expected to replace laser irradiation to trigger extravascular gelation after replacing GNRs with $Fe_3O_4$, which can treat tumors in deeper tissues, since magnetic trigger features deeper penetration than light[23].

In summary, we established a starvation therapy method based on the vascular blockage by extravascular gelation shrinkage-derived internal stress for squeezing blood vessels. An organic-inorganic composite hydrogel (hydrogel–GNR) consisting of GNRs and CS/mPEG-Mal/pNIPAAm-co-AAc with a LCST at 39 °C has been constructed to validate it. Such a hybrid hydrogel can shrinkage upon exposed to 808 nm laser irradiation and produce internal stress to squeeze and narrow blood vessels, ultimately occluding ex vivo and in vivo blood & nutrition supply, which enables in vivo regression of PANC-1 pancreatic cancer on nude mice with prolonged survival rate. Similar results and conclusions were also obtained on 4T1-bearing BALB/c mice, which demonstrate the generality of this starvation therapy. More

significantly, the decreased pore diameter and porosity in hydrogel–GNR can effectively impede cell migration and invasiveness, thus serving as a countermeasure for successfully inhibiting tumor metastasis on transgenic mice-bearing orthotopic breast cancer and nude mice-bearing PANC-1 pancreatic cancer, respectively. As well, it can act as an assisted method to inhibit residual tumor recurrence after photothermal ablation. Such an extravascular starvation therapy holds tremendous potential in addressing the three primary concerns, i.e., tumor progression, tumor recurrence and tumor metastasis.

## Methods

**Lower critical solution temperature (LCST) study.** LCSTs of the pNIPAAm-*co*-AAc hydrogel with tunable AAc ratios were determined from the intersection where 50% absorption and 50% viscosity were observed. The temperature-dependent transmission of the precursor solution at 450 nm was continuously recorded at a heating rate of $1.0 \, °C \, min^{-1}$ by a Shimadzu 3101PC UV/Vis spectrophotometer equipped with a peltier temperature control accessory. The temperature-dependent viscosity of the precursor solution was determined at a heating rate of $1.0 \, °C \, min^{-1}$ using DV-T1 digital viscometer system (Shanghai Nirun Intelligent Technology Co., Ltd) equipped with programmed temperature elevation unit. The temperature-dependent measurements regarded room temperature (20 °C) as the starting point. Furthermore, an appropriate AAc content corresponding to an optimal LCST was chosen to investigate the temperature-responsive gelation process of chitosan/mPEG-mal/pNIPAAm-co-AAc composite hydrogel and its combined hydrogel with GNRs (namely hydrogel–GNR), especially referring to the temperature-dependent viscosity and absorption at 450 nm.

**In vitro temperature variations exposure to 808 nm laser.** 3.5 mL of above-mixed aqueous precursor solution was transferred into a resin cuvette with four transparent sides, which had a diameter of 1 cm and a depth of 4 cm. A thin cover glass was covered on the cuvette to prevent the solution from drying during the experiments. The power density and concentration-dependent photothermal transitions were evaluated on an 808-nm continuous wave (CW) laser (Shanghai Institute of Optics and Fine Mechanics, Chinese Academy of Sciences) with tunable power density at $0–1.0 \, W \, cm^{-2}$. The quantitative temperature variation and visual thermal infrared images during the laser irradiation were recorded on digital temperature recording system with a thermoelectric couple and an IR camera (FLIR SC300-Series), respectively. Photothermal-induced gelation experiments were performed in a constant temperature & humidity incubator (25 °C) during laser irradiation. After the irradiation, DI water was added to the solution-containing cuvette to evaluate gel formation, and the gel was considered to form only if an insoluble composition was found. The optimal irradiation time and power density was determined as the starting point where gel formation occurred.

**Rheological characterization.** To further determine the gelation temperature, injectability and gelation time, the rheological experiments were performed on a Thermo Haake Rheostress RS6000 rheometer (Thermo Scientific, Karlsruhe, Germany) with parallel-plate geometry (with diameter of 20 mm and a gap of 1 mm). The relationships of shearing rate with shearing stress and viscosity were explored, respectively, to evaluate the injectability of this composite hydrogel–GNR or pNIPAAm. To understand the gelation kinetics, the storage modulus ($G'$) and loss modulus ($G''$) of hydrogel and hydrogel–GNR were monitored as a function of time and temperature, respectively. The dynamic time sweep was carried out at a fixed strain of 1% and a constant frequency of $1 \, rad \, s^{-1}$. Their temperature-dependent rheological properties were then monitored at 1 Hz (frequency) and 1 Pa (stress) over a continuous linear temperature ramp at $1 \, °C \, min^{-1}$. To further investigate the elastic and viscous properties of hydrogel and hydrogel–GNR, the frequency-dependent and strain-dependent sweep tests were carried out. The frequency sweep curves were carried out at a fixed strain of 1% with an angular frequency range from 1 to $100 \, rad \, s^{-1}$. The dynamic oscillatory strain sweep of gels was carried out at a fixed frequency of 1 Hz. The measurements for each sample were performed in triplicate.

**Mechanical measurement.** The tensile stress–strain measurements of pNIPAAm, composite hydrogel and composite hydrogel–GNR were performed on gels using a tensile/compressive tester (FR-108B, Farui Co., China). The cylindrical gel samples with 13 mm in diameter and 7 mm in thickness were set on the lower plate and stretched by the upper plate which was connected to a load cell (500 N). The compressive tests of the gels were obtained at the speed of $1 \, mm \, min^{-1}$. The tensile stress ($\sigma$) was approximately calculated as $\sigma = F_{load}/\pi R^2$, where $R$ is the original radius of the specimen and $F_{load}$ indicate the applied force. The tensile strain ($\varepsilon$) is defined as the increment magnitude of the thickness relative to the original thickness. The stress and strain between $\varepsilon = 5$ and 15% were used to calculate the elastic modulus with at least parallel tests for each hydrogel. Noticeably, every test was repeated for three times.

**In vitro biosafety evaluation.** Murine fibroblast cell (L929 cells, catalogue number: GNM28) was obtained from Shanghai Institutes of Biological Sciences, Chinese Academy of Sciences, and was cultured in high glucose Dulbecco's Modified Eagle's Medium (DMEM) containing 10% fetal bovine serum and 1% penicillin/streptomycin. The cells were cultured at 37 °C in a humidified atmosphere with 5% $CO_2$, and harvested with 0.25% trypsin-EDTA and rinsed. The cell suspension obtained was used in the following experiments. The 3-(4,5-dimethylthiazol-2-yl)-2,5-diphenyltetrazolium bromide (MTT)-based cytotoxicity assay was performed to evaluate the cellular response to residual solution of different concentrations. The mixture solution-containing residual precursor solution and DMEM culturing medium with a tunable ratio was added into the 24-well plate (Costar, Corning Incorporated, Corning, NY, USA) containing $8 \times 10^4$ cells per well and incubated for 24 h. Afterwards, the culturing mixture was removed and washed with PBS for 3 times, and then 400 μL of MTT ($8 \, mg \, mL^{-1}$ stock) solution was added in each well, and the cells were further incubated for 4 h. Immediately afterwards, the MTT solution was sucked out, and 400 μL of DMSO solution was added, and furthermore, the fluorescence intensity at 480 nm was read on a fluorospectrophotometer (F-2500, Hitachi, Japan) with an excitation wavelength of 480 nm.

**Ex vivo blood vessel squeezing and occlusion test.** SD rattus norvegicus were supplied by Laboratory Animals Center of Tenth Peoples' Hospital of Tongji University, and were kept in sterilized cages with supply of filtered air, sterile food, and water. The experiments were performed according to protocols approved by the Laboratory Animal Center of Shanghai Tenth Peoples' Hospital and were in accordance with the policies of National Ministry of Health. After sacrificing, two aorta vessels with a length of 5 cm and an inner diameter of 1.4 mm were stripped out and rinsed by PBS for use. Their middle sections were immersed into hydrogel–GNR solution and hydrogel–GNR gel, respectively, and one tail end was linked to controllable syringe pump whose rate was set $72 \, mL \, h^{-1}$ so as to guarantee ($1.3 \, cm \, s^{-1}$) that the calculated $1.3 \, cm \, s^{-1}$ in rat aorta was approximately identical to the real flowing rate in melanoma tumor. Red DMEM and blood with heparin anticoagulation was employed model blood for monitoring the flowing rate, respectively. The whole experimental process was recorded by camera and the collected DMEM or blood out of another tailed end of aorta was photographed.

**In vitro inhibitory migration experiment.** The special culturing dishes available for investigating the inhibitory migration behavior of hydrogel–GNR against human pancreatic cancer cells, i.e., PANC-1 cells (catalogue number: SCSP-535, Shanghai Institutes of Biological Sciences, Chinese Academy of Sciences) were first designed and manufactured. In detail, normal culturing dishes were washed with ethanol and DI water to remove oil contamination. Subsequently, hydrogel–GNR solution ($5 \, g \, mL^{-1}$, 2 mL) was added and heated to induce gelation. Subsequently, five square glass slides (1.5 cm × 1.5 cm) were placed and adsorbed on the hydrogel–GNR gel via the capillary force, and they were allocated in a cross spread pattern. Furthermore, 10 mL of aqueous hydrogel–GNR solution was added in the culturing dishes, and heated to induce gelation. Finally, the hydrogel–GNR gel vertical to square glass slide was cut off and generate five culturing tanks, ultimately obtaining the special culturing dishes. PANC-1 cell dispersion in DMEM-based culturing medium (density: $2 \times 10^5$ per mL, volume: 3 mL) were seeded in the middle culturing tank (named as S0), and pure EMDM-based culturing medium (3 mL) was added into other four culturing tanks (named as S1-S4) as control. At certain time points, the number of PANC-1 cell in each culturing tanks recorded.

**In vitro degradation of hydrogel–GNR.** Configuration of SBF solution was carried out according to previous report. Typically, First, 7.996 g of NaCl, 0.350 g of $NaHCO_3$, 0.224 g of KCl, 0.228 g of $K_2HPO_4\cdot3H_2O$, 0.305 g of $MgCl_2\cdot6H_2O$ and 20 mL of 2 M hydrochloric acid were dissolved into 750 mL of DI water, then 0.278 g of $CaCl_2$, 0.071 g of $Na_2SO_4$, and 6.057 g of tris(hydroxymethyl) aminomethane were slowly added in turn, finally the fluid was buffered to 1 L at physiological pH 7.4 at 37 °C with dilute hydrochloric acid solution. 10 mg hydrogel–GNR was immersed into 100 mL of SBF solution. At some certain points, five fresh and independent hydrogels were taken out and rinsed by deionized water for 3 times, followed by drying under vacuum and weighing, and the lost weight was recorded and plotted.

**In vivo tests on degradation and biosafety.** Kunming mice with an average body weight of about 20 g were supplied by Laboratory Animals Center of Tenth Peoples' Hospital of Tongji University, and the experiments were performed according to protocols approved by the Laboratory Animal Center of Shanghai Tenth Peoples' Hospital and were in accordance with the policies of National Ministry of Health. Four groups with 6 nude mice in each group were randomly chosen before experiments, i.e., control and experimental group. Ellipsoidal hydrogel–GNR blocks (1.8 cm × 1.5 cm) were subcutaneously embedded in the back of Kunming mice in the experimental group. The gel volume was measured in two dimensions at certain time point by using an electronic caliper and calculated using the following formula: Volume = length × width$^2$/2. The degradation of hydrogel-GNR was obtained according to the volume variation profile of hydrogel–GNR gel. On day 0 and day 45, mice were sacrificed, and blood and other tissues (heart, liver, pancreas, spleen, lung, brain, stomach and kidney) were harvested. Blood and

biochemical indices were obtained to evaluate the blood safety, while these normal tissues were stained by Hematoxylin and Eeosin (H&E) to evaluate the tissue biosafety of this hydrogel–GNR.

**In vivo vascular occlusion test on blood vessels in abdomen**. Kunming mice with an average body weight of about 20 g were supplied by Laboratory Animals Center of Tenth Peoples' Hospital of Tongji University, and the experiments were performed according to protocols approved by the Laboratory Animal Center of Shanghai Tenth Peoples' Hospital and were in accordance with the policies of National Ministry of Health. 2 mL of hydrogel–GNR precursors was intratumorally (i.t.) injected into the periphery of vein blood vessels in abdomen under the guidance of CDFI on SIEMENS S2000 (probe: 18L6 HD), and then laser irradiation was carried out to trigger the gelation and shrinkage for generating internal stress. Before and after administering, color Doppler ultrasonography images and movies were captured and recorded.

**Model establishment of PANC-1 and 4T1 xenografted tumor**. PANC-1 cells were cultured in DMEM containing 10% heat-inactivated fetal bovine serum and 1% antibiotics (streptomycin and penicillin) (Invitrogen). The culturing dishes were maintained in a humidified atmosphere with 5% $CO_2$ at 37 °C. Nude mice (Balb/C nu/nu) with an average body weight of about 20 g were supplied by Laboratory Animals Center of Tenth Peoples' Hospital of Tongji University, and were kept in sterilized cages with supply of filtered air, sterile food, and water. During establishing the solid tumor model, 0.2 mL cell suspension ($2 \times 10^6$ cells) in DMEM medium without serum/phenol red, was injected subcutaneously into the back of nude mice using a 27-gauge needle. Tumors were allowed to grow to $100 \pm 20$ mm$^3$ (mean ± standard deviation) prior to any experiment (designated as experimental day 1). Mice were monitored for general well-being, weight, tumor volume and survival rate.

Similar procedures are available for establishing 4T1 breast xenografted tumor wherein nude mice and PANC-1 cells in PANC-1 model establishment were replaced by immune-competent Balb/C mice (female) and 4T1 cells, respectively. 4T1 cell line (catalogue number: SCSP-535) was purchased from Shanghai Institutes of Biological Sciences, Chinese Academy of Sciences. All in vivo experiments were performed according to protocols approved by the Laboratory Animal Center of Shanghai Tenth Peoples' Hospital and were in accordance with the policies of National Ministry of Health.

**In vivo photothermal temperature measurements**. 200 μL of hydrogel–GNR solution in PBS (5 g per 100 mL) was i.t. injected into the PANC-1 tumor implanted on nude mice ($n = 3$). Subsequently, 808 nm laser irradiation at a power density of 0.15 W·cm$^{-2}$ was carried out, during which the time-dependent temperature variation was visually monitored and recorded on the IR camera (FLIR SC300-Series) to determine the optimal irradiation duration.

**In vivo starvation therapy against PANC-1 and 4T1 tumors**. Five experimental groups were randomly chosen before experiments ($n = 6$), i.e., control group (designated as G1) that only saline was i.t. injected into tumor; Lase+GNR group (designated as G2) wherein 808 nm laser irradiation was applied after i.t. injecting GNRs into tumor; Laser+hydrogel–GNR (1) group (designated as G3) wherein 808 nm laser irradiation was applied after i.t. injecting hydrogel into tumor alone, Laser+hydrogel–GNR (2) group (designated as G4) wherein 808 nm laser irradiation was applied after i.t. injecting hydrogel–GNRs into both tumor and its surrounding periphery, and pNIPAAm group (designated as G5) wherein only pNIPAAM was i.t. injected into tumor and its periphery that spontaneously trigger gelation due to the 32 °C of LCST by the body temperature of living body. The concentration of hydrogel or hydrogel–GNR (20 ppm) was 5 g per 100 mL, and the injection volume was 200 μL. In addition, to avoid body temperature drop during anesthesia and minimize experimental variation, the nude mice-bearing PANC-1 tumors or BALB/c bearing 4T1 tumors were placed on 37 °C constant plate. At the beginning of experiment, 200 μL of each solution were locally injected into the solid tumor using a multisite intratumoral injection subcutaneously from the tumor edge, which benefited homogeneous distribution of hydrogel precursors within the tumor tissue compared to single-site injection. More significantly, an identical treatment in each group was also conducted in the periphery of solid tumor. After injection, pulsed irradiation using an 808 nm laser at a power density of 0.15 W cm$^{-2}$ was instantly implemented, and the duration was 3 min in sum with pulse duration of 30 s and interval time of 10 s, during which thermal infrared scanning on IR camera (FLIR SC300-Series) was used to monitor the temperature and less than 42 °C should be guaranteed so as to avoid thermal injures.

The survival rate and body weight of nude mice or BALB/c mice in each groups were recorded at certain time points, during which the tumor size was simultaneously measured in two dimensions (length and width) using an electronic caliper and its volume was calculated according to the following formula: Tumor volume = length × width$^2$/2. Especially in the group of Laser+hydrogel–GNR (2) (i.e., G4), the blood perfusion in tumor before and after laser irradiation was recorded via the CDFI technology on Toshiba Aplio500 (probe: 14L5) so as to directly evaluate the in vivo blood occlusion. In detail, CDFI were carried out to capture images and record movies of blood velocity variation in tumor tissues

before and after the corresponding treatment in Laser+hydrogel–GNR (2), and on 5th day post-treatment, the images and movie of blood velocity variation in tumor tissues was further recorded.

On the 10th day, some mice in the five groups ($n = 3$) were sacrificed and their tumors were taken out for various pathological examinations including western blot, CD34 & TUNEL immunofluorescence co-staining, hematoxylin and eosin (H&E), CD34 and proliferating cell nuclear antigen (PCNA) immumohistochemical stainings were carried out, and then observed via laser confocal scanning microscopy (LCSM) on Nikon A1R/A1. The semi-quantitative number and percentage of apoptotic cells in total, apoptotic tumor cells and apoptotic endothelial cells labeled by CD34 can be obtained via counting the percentage of LCSM images in CD34 & TUNEL immunofluorescence co-staining. All uncropped western blot scans have been provided in the Source Data file.

More significantly, objective to 4T1 tumor model implanted on immune-competent BALB/c mice, the robust influences of such special treatment (G4) on inflammatory milieu in tumor has been evaluated. In detail, on the 10th day, tumors were excised for analysis by flow cytmetry (FCM) after co-staining with anti-CD11b FITC, anti-CD45 BV510, anti-F480 APC, and anti-CD11c PE for differentiating lymphocytes, macrophages and DC cells. Furthermore, DC cells were analyzed by FCM after co-staining with anti-CD45 BV510, anti-CD3 FITC, and anti-CD49b APC for differentiating natural killing cells and T-lymphocytes. Moreover, T-lymphocytes can be divided after analysis by FCM after co-staining with anti-CD4 PE-Cy7 and anti-CD8 BV421. Meanwhile, the inflammation-related cytokines in harvested tumor tissue samples were tested by enzyme-linked immunosorbent assay (ELISA).

**Detailed information of all used antibodies**. The names, suppliers and catalog numbers of employed antibodies for western blotting, immunofluorescence, immunohistochemistry, flow cytometry and ELISA are given in the follow part and they are listed as antigen first, followed by supplier, catalog number and clone/lot number as applicable. All operations referred to the instructions and no dilution was carried out. In western blotting assay, Anti-bcl-2 antibody, Abcam, cat. no. ab32124; Anti caspase-3 antibody, Abcam, cat. no. ab13847; Anti-HSP70 antibody, Abcam, cat. no. ab2787; Anti-p53 antibody, Abcam, cat. no. ab131442; Anti-CD34 antibody, Abcam, cat. no. ab8158; Anti-CD31 antibody, Abcam, cat. no. ab28364; Anti-beta Actin Antibody, Abcam, cat. no. ab6276; Anti-HIF-1 alpha antibody, Abcam, cat. no. ab16066. In immunofluorescence staining, Anti-mouse CD34, Abcam, cat. no. ab8158; Anti-mouse CD31, Abcam, cat. no. ab24590; DAPI, Beyotime Biotechnology, cat. no. C1002; TUNEL apoptotic assay kit, Roche Applied Science, cat. no. 11684817910. In immunohistochemical staining, Anti-mouse PCNA, Abcam, cat. no. ab29; Anti-CD34 antibody, Abcam, cat. no. ab8158; Hematoxylin and Eosin Staining Kit, Beyotime Biotechnology, cat. no. C0105. In flow cytometry analysis, Anti-mouse CD45, Biolegend, cat. no. 103108, Clone: 30-F11; Anti-mouse CD11b, Biolegend, cat. no. 101208, Clone: M1/70; Anti-mouse F4/80, Biolegend, cat. no. 123116, Clone:BM8; Anti-mouse CD49b Antibody, cat. no. 103516, Clone: HMα2; Anti-mouse CD3, Biolegend, cat. no. 100204, Clone: 17A2; Anti-mouse CD4, Biolegend, cat. no. 100412, Clone: GK1.5; Anti-mouse CD8, Biolegend, cat. no. 140408, Clone: 53-5.8; Anti-mouse CD11c, Biolegend, cat. no. 117310, Clone: N418. In ELISA assay, Ani-mouse IL-1β, Invitrogen, at. no. MM425B; Anti-mouse IL-4, eBioscience, cat. no. 14-7041-81; Anti-mouse IL-5, Invitrogen, cat. no. MM550CB; Anti-mouse IL-13, eBioscience, cat. no. 14-7133-81; Anti-mouse IL-17, eBioscience, cat. no. 13-7177-81; Anti-mouse IL-18, Invitrogen, cat. no. PA5-81413; Anti-rat IL-21, Invitrogen, cat. no. MA5-30812; Anti-mouse IL-23, eBioscience, cat. no. 13-7123-81; Anti-mouse IL-6, eBioscience, cat. no. 14-7061-81; Anti-mouse IFN gamma, Invitrogen, cat. no. BMS606INST; Anti-mouse TNF α, eBioscience, cat. no. 14-7325-81; Anti-mouse TNF β, eBioscience, cat. no. BMS105; Anti-mouse IL-8, Invitrogen, cat. no. M801; Anti-mouse IL-12, eBioscience, cat. no. 16-7101-81; Anti-mouse CCL-22, Invitrogen, cat. no. MA5-23780; Anti-mouse CCL-17, Invitrogen, cat. no. PA5-78933; Anti-mouse CCL-5, eBioscience, cat. no. 14-7085-82.

**In vivo metabolism test**. Nude mice-bearing PANC-1 tumor ($n = 3$) was employed, and GNR metabolism was used to evaluate the degradation and secretion of this composite hydrogel, since robust chemical chelation between GNR and pNIPAAm-co-AAc can withstand erosion and no Au atoms in living body. After treatment with G4 (i.e., Laser+hydrogel–GNR (2)), nude mice were kept in special metabolic cage, and at each certain time point, faeces and urine were collected to be dissolved by nitrohydrochloric acid under heating at 50 °C. After complete dissolution, undissolved residues were discarded via filtration by 200 nm filtering membrane, and the dissolved solution was normalized to a constant volume (10 mL) by adding ultrapure water. Ultimately, Au concentration in the normalized solution was determined by inductively coupled plasma optical emission spectrometry (ICP-OES) on Agilent Technologies 725, and the secreted Au mass can be obtained. Ultimately, the time-dependent accumulatively secreted percentage of Au-labeled hydrogel–GNR can be obtained via calculating the ratio of accumulative Au mass to initially added Au mass.

**Indirect intratumoral vascular occlusion**. Two groups were randomly chosen before experiments, i.e., Laser+GNR group (i.e., control group, designated as G2)

wherein 808 nm laser irradiation was applied after i.t. injecting GNRs into both tumor, and Laser+hydrogel–GNR (2) group (i.e., treated group, designated as G4) wherein 808 nm laser irradiation was applied after i.t. injecting hydrogel–GNRs into both tumor and its surrounding periphery. First, fluorescence imaging was used to evaluate the intratumoral vascular occlusion arising from the squeezing and narrowing of intratumoral blood vessels caused by this extravascular gelation shrinkage-induced internal stress. In detail, after corresponding treatment in each group, nude mice-bearing PANC-1 tumor or BALB/c mice-bearing 4T1 tumor in each group ($n = 3$) was instantly chosen, and FITC-labeled dextrans with varied sizes (i.e., 70, 20, and 4 K) were intravenously (i.v.) injected into nude mice or BALB/c mice in both groups via tail vein. The in vivo animal fluorescence imaging on IVIS Lumina II with Andor iKon camera was carried out to capture the fluorescence images after 10 min.

Subsequently, ultrasound contrast imaging technology under CHI via injecting microbubbles was also used to indirectly evaluate the intratumoral vascular occlusion. Nude mice-bearing PANC-1 tumor or BALB/c mice-bearing 4T1 tumor in each group ($n = 3$) was chosen. Before corresponding treatment in each group, the 1st i.v. injection of clinically used Sonovue$^{TM}$ microbubbles (0.2 mL) into nude mice or BALB/c mice was carried out in both groups via tail vein, and then the ultrasonic images and movies under both CHI and B fundamental imaging (BFI) modes was instantly captured on GE LOGIQ E9. After 30 min-90 min, the corresponding treatments in both groups, i.e., treated (i.e., Laser+hydrogel–GNR or G4) and control (i.e., Laser+GNR or G2) groups, were carried out. Afterwards, the 2nd i.v. injection of Sonovue$^{TM}$ microbubbles into nude mice or BALB/c mice was implemented in both groups via tail vein, and then the ultrasonic images and movies under CHI mode was instantly captured again. As well, the occlusion test of another group, i.e., pNIPAAM group (G5), was also evaluated on nude mice-bearing PANC-1 tumor using in vivo fluorescence imaging (FITC-labeled dextran (4 K) as tracers) and ultrasound contrast imaging (Sonovue$^{TM}$ microbubbles as tracers), respectively.

Ultimately, nude mice-bearing PANC-1 xenografted tumor ($n = 3$) was used to evaluate the diffusion of composite hydrogel, wherein Cy3-labeled dextran (size: 70 K) was first used instead of GNRs to label the composite hydrogel, and was i.t. injected into PANC-1 tumor, and in vivo animal fluorescence imaging was carried out on IVIS Lumina II with Andor iKon camera at three time points, pre, 30 s post- and 50 s post-i.t. injection of hydrogel–dextran into PANC-1 tumor. Afterwards, T2-weighted MRI under the T2-TSE-Tra sequence on SIEMENS MAGNETOM Verio syngo MR B17 was carried out instead of in vivo animal fluorescence imaging for evaluating the diffusion of hydrogel–GNR, and hollow porous $Fe_3O_4$ nanoparticles act as the T2 contrast tracer. Before and after treatment with injection of contrast agents, T2-weighted MR images were captured using the following parameters: TR = 1740 ms, TE1 = 4.36 ms, TE2 = 11.90 ms, TE3 = 19.44 ms, TE4 = 26.98 ms, TE5 = 34.52 ms, BW = 260 Hz, Thickness = 2 mm, Slices = 20, FOV = 200*200.

As well, after 1 day post-corresponding treatments in both groups including Laser+GNR group (G2, designated as control group) and Laser+hydrogel–GNR (2) (G4, designated treated group), nude mice-bearing PANC-1 tumors in each group ($n = 3$) was instantly chosen and sacrificed and tumors were excised. Afterwards, Cy 3-labeled CD31 and FITC-labeled CD34 immunofluorescence stainings of tumor tissue slices were instantly carried out in two groups for LCSM observation on Leica TCS SP5.

**In vivo test on inhibiting tumor skin–skin metastasis.** Two groups with 6 mice-bearing PANC-1 tumor in each group were set, and they were control and experimental group, respectively. In control group (i.e., US(MBs)), microbubbles were first injected, followed by ultrasound irradiation, US irradiation was used to trigger tumor metastasis. In contrast, in the experimental group (Laser+hydrogel–GNR+US (MBs)), hydrogel–GNR was first injected into the tumor tissue and its surrounding periphery, and then generated the gelation process via laser irradiation, which was identical to the treatment in aforementioned Laser+hydrogel–GNR (2). Subsequently, the treatment that is marriage of intravenous MBs injection and US irradiation was carried out. Laser irradiation with a power density of 0.15 W cm$^{-2}$ was instantly implemented after injecting hydrogel–GNR, and the duration time was 3 min in sum with pulse duration of 30 s and interval time of 10 s. The concentration of hydrogel–GNR (20 ppm) was 5 g per 100 mL, and the injection volume was 200 μL. Plane ultrasound irradiation that was carried out on a portable ultrasound apparatus (Chattanooga, USA) at the power of 1 W (100% duty cycle) was also pulsed with four cycles, and each cycle contained a 30 s irradiation duration and an interval time of 30 s between two irradiations, and the ultrasound irradiation under the same conditions was repeated for 5 days at the site of the primary tumor. The growth process of PANC-1 tumor was monitored in a real-time manner, and the metastasis site of PANC-1 tumors was recorded after 10 days.

**In vivo inhibition test of tumor recurrence.** Photothermal resection was used to investigate the tumor recurrence. Two groups with 10 nude mice-bearing PANC-1 tumor in each group were set, and they were designated as control group and experimental group, respectively. In the experimental group, hydrogel–GNR first injected into the periphery of PANC-1 tumor, and laser (1) was applied to induce the gelation and make gel wrap the PANC-1 tumor. Before and after treatment, CDFI on SIEMENS S2000 (probe: 18L6 HD) was used to monitor the variation of blood vessels around the tumor. Afterwards, GNRs were i.t. administered into the

PANC-1 tumor, and laser (2) was enforced. In the control group, GNRs were directly administered into the PANC-1 tumor, followed by treatment with laser (2) for ablating the tumor. Herein, intratumoral injection of GNRs into PANC-1 tumor is expected to incompletely ablate tumor in the presence of laser (2) irradiation and generate residual tumor tissues available for recurrence, which is one typical model of tumor recurrence. In treated group, the pre-treatment with injection of hydrogel–GNR into the normal tissues surrounding PANC-1 tumor and laser (1) irradiation is anticipated to trigger gelation process of such a composite hydrogel, narrow periphery blood vessels surrounding tumor and occlude blood and nutrition supply towards residual tumor tissues and ultimately suppress tumor recurrence. Herein, laser (1) irradiation with a power density of 0.15 W cm$^{-2}$ was instantly implemented after injecting hydrogel–GNR, and the duration time was 3 min in sum with a pulse duration of 30 s and an interval time of 10 s, and the concentration and volume of hydrogel–GNR (3%) were 5 g per 100 mL and 500 μL. The parameters of laser (2) irradiation were power density of 0.30 W cm$^{-2}$ for continuous 5 min, and the dose and volume of GNRs were 10 nM and 200 μL. The tumor recurrence in two groups was monitored after 5 days.

**In vivo test on inhibiting breast-to-lung metastasis.** A transgenic mouse model that can generate orthotopic mammary adenocarcinomas after induction by MMTV-PyVmT oncogene has been accepted and used as the ideal model for evaluating anti-metastasis basing on this special treatment in this revised manuscript, as the majority of the tumor-bearing transgenic mice develop secondary metastatic tumors in the lung, as evidenced by many references. The transgenic mice with an average body weight of about 20 g were supplied by Laboratory Animals Center of Tenth Peoples' Hospital of Tongji University that purchased from Shanghai Hanyin biotech. Co.LTD, and the experiments were performed according to protocols approved by the Laboratory Animal Center of Shanghai Tenth Peoples' Hospital and were in accordance with the policies of National Ministry of Health. Transgenic mice-bearing orthotopic breast tumor were randomly divided into two groups ($n = 7$), i.e., control (G1) and treated (i.e., G4 or Laser+hydrogel–GNR (2)) groups. After corresponding treatment in each group, transgenic mice were raised, and at the end day of experiments (day 28), the mice in both groups were sacrificed and lungs were excised and stained by triphenyltetrazolium chloride (TTC) according to the instruction. Eye-naked observation was carried out to count the number of tumor transferred from the primary tumor in subcutaneous breast pad.

**Statistical analysis.** All the experiments were performed in triplicate. The obtained data were expressed as the mean value ± standard deviation (SD) and the statistical significance between two groups was analyzed by unpaired Student's $t$ test through SPSS 19.0 (SPSS, Chicago, IL, USA). Single, double and triple asterisks represent $P \leq 0.05$, 0.01, and 0.001, respectively, wherein $*P < 0.05$ was considered statistically significant and $**P < 0.01$ was extremely significant.

**Reporting summary.** Further information on research design is available in the Nature Research Reporting Summary linked to this article.

## Data availability
All relevant data are available from the correspondence authors upon reasonable request. The source data including Figs. 1c, 3c-f, 4c,e, 5b,c,g, 6b,c, 7b,c,e,g-i and 8c,d,h and Supplementary Figs 9, 16, 17, 18, 20, 22, 25, 26, 27, 29, 30b and 31 are provided as a Source Data file, and they also have been uploaded in a generalist repository named Open Science Framework with an identifier (https://doi.org/10.17605/OSF.IO/WQF7A).

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

## Acknowledgements

This work was supported by grants from National Natural Science Foundation of China (Grant No. 81771836, 81501473, 81671695, 81601502, and 81725008), the Fostering Project of Shanghai Municipal Commission of Health and Family Planning for Excellent Young Medical Scholars (Grant No. 2018YQ31), the Shanghai Science and Technology Committee Rising-Star Program (A type) (Grant No. 19QA1406800), the Opening

Project of Guangxi Key Laboratory of Bio-targeting Theranostics (Grant No. GXSWBX201801), and the Opening Project of State Key Laboratory of High Performance Ceramics and Superfine Microstructure (Grant No. SKL201811SIC). We thank Xia Wang and Qigang Wang at school of Chemical Science and engineering (Tongji University) for providing apparatus of rheological characterization and mechanical measurement, as well as Ning Wang at Shanghai Institute of Ceramics, Chinese Academy of Sciences for advice and technical assistance.

## Author contributions

K.Z. conceived and designed the project. K.Z., Y.F., Y.H., H.Y., X.G., Y.P., B..Z., W.Y., and H.L. performed in vitro experiments, and K.Z., D.D., Y.F., Y.H., W.R., C.L., and L.S. performed in vivo experiments. K.Z. wrote the manuscript, and K.Z., Y.C., and H.X. supervised the project. All authors discussed the results and commented on the manuscript.

## Competing interests

The authors declare no competing interests.
