## [Peer Review File · Nature Communications]

Reviewers' comments:

Reviewer #1 (Remarks to the Author):

Summary: The authors describe a novel and elegant approach to efficiently target vasculature as a means of controlling primary tumor growth, recurrence and metastasis—the bane of current anti-tumor therapies—with minimal off-target or toxic effects. While targeting vasculature, this approach does not rely on vascular rupture or embolism, two approaches with known limitations. Thus, the reported technology is an advancement on existing methods. The paper is well-written. The graphic schematics that help explain the experimental set-ups, as well as the detailed figure legends, were welcome and well-executed. The animal studies are sufficiently powered as well.

However, below are some concerns I would like the authors to address prior to acceptance of their work.

- 1 - For their in vitro studies of tumor invasion, Fig. 5g and h, the quantification shows that the cell number continues to increase over time in the other chambers (S1-4). Is this due to proliferation or influx of cells from S0? In which case, the hydrogel GNR is not inhibiting? It would be interesting to see other hydrogels as control in this experiment to gauge the potency of the test material.
- 2 - Figure resolutions, especially where images are shown such as in Fig. 4d-h, is very low. Please provide higher magnification and higher resolution images for the reviewers.
- 3 - Please provide an insight into the cells that are TUNEL positive in Fig.4d. Is it primarily the cancer cells or do the host cells also die? I suggest co-staining with a pancreatic epithelial cell marker at the minimum. This is important given the importance of the host cells in the tumor microenvironment for promoting primary tumor recurrence and metastasis. Also, I would like to see a syngeneic model in immune competent mice to get a sense of how robust the treatment is in overcoming the inflammatory milieu relevant to the solid tumor.
- 4 - The authors show ex vivo using intact blood vessel that the gel is able to block flow. How does this translate in vivo, especially in the tumors where the vasculature is leaky? I suggest in vivo perfusion studies with labeled dextran of varying sizes to test this. Would delivery of dextran to the tumors be limited due to constricted vessels? This would be an important and relevant study to do.
- 5 - Related to apoptosis that is evident in tumors: Fig. 4B shows a cytostatic effect of treatment and not a cytotoxic effect. How does this tie in with the apoptosis data? Another reason why it is important to identify which cells are undergoing apoptosis (percent of tumor and cell type), since it appears that this level is not sufficient to shrink the tumor.
- 6 - A conceptual question that should be addressed in the discussion: Since the tumor is present after treatment, secondary treatment may be required such as chemotherapy. However, how do the authors suggest delivery of the chemotoxic agent if the gel-based treatment constricts

the vasculature? At what stage of tumor development would this treatment be most effective? For early stage tumors surgical removal is still the accepted strategy. For late stage tumors this therapy may not be potent enough to act in a timely manner. Another concern is since the in vivo life is about 60 days. Will repeat procedures be required? This could counteract the anti-metastatic effect of the treatment. Looking at additional tumor models will increase the robustness by shedding light on applicability of treatment in a relevant context.

With a bit of additional data and more clarity, I recommend reconsideration for positive action and acceptance to Nature Communications.

Reviewer #2 (Remarks to the Author):

This paper describes a gold nanorod and thermally-sensitive hydrogel mixture that undergoes gelation shrinkage. The authors use this gel to enable vascular shutdown on injection into tumor. The hypothesis is that the gelation causes the constriction of the blood supply to the tumor, which then results in suppression of metastasis and recurrence. This is a simplistic theory, but there are many questions.

1. the gel is being injected into the tumor and allowed to stiffen. Although the authors perform ex vivo studies to show that this gelation blocks flow through a vessel, this has no relevance to a tumor microenvironment. For example, a major challenge in the tumor environment is diffusibility- does the gel diffuse uniformly through the tumor (could be tested using antibodies raised against the polymer or using tracers). A drop in blood flow in the tumor could just have been due to increased intratumoral pressure as a result of the injection of the hydrogel in the tumor.

2. What is strange is the reduction in biomarkers of vasculature following injection. If the mechanism is via occlusion, the biomarker levels should not change. The levels should only change if an agent that prevents angiogenesis (or causes the destruction of endothelial cells) is administered. Histology images in Fig.4 are not convincing of the claims made.

3. One should have expected occlusion to increase HIF1a. Instead, the authors claim an increase in P53. How is that happening?

4. In Fig. 5, the tumor is shown to metastasize from one sub-cutaneous site of injection to another subcutaneous site. The authors claim these sub-cut sites as distant metastasis sites. In my 25 years of working in cancer biology, I am yet to see such sub.cutaneous to sub.cutaneous metastasis. Are the authors sure that the cells were not injected into the other sub. cutaneous sites by mistake? Typically, one would see mets to distant organs.

5. The authors claim that this approach avoids intravascular aggregates-induced embolisms. Given that the hydrogel disappears after a period of time, one needs to test where it goes, and degradation products in blood could have limitations as current intravascular approaches. If the hydrogel degrades over time, why is recurrence inhibited?

6. Minor points:

a. Please use exact numbers instead of 'some mice were sacrificed'. Provide n values for every study in legend, together with statistical tests performed.

b. In some sentences, parts are missing; for example, line 102- I presume LCST of the pure polymer is 32C.

Response to Reviewer #1

Reviewer #1 (Remarks to the Author):

The authors describe a novel and elegant approach to efficiently target vasculature as a means of controlling primary tumor growth, recurrence and metastasis—the bane of current anti-tumor therapies—with minimal off-target or toxic effects. While targeting vasculature, this approach does not rely on vascular rupture or embolism, two approaches with known limitations. Thus, the reported technology is an advancement on existing methods. The paper is well-written. The graphic schematics that help explain the experimental set-ups, as well as the detailed figure legends, were welcome and well-executed. The animal studies are sufficiently powered as well.

Response: Thank you very much for the positive comments and constructive suggestions.

Please find the following detailed responses to your comments and suggestions.

However, below are some concerns I would like the authors to address prior to acceptance of their work.

1 - For their *in vitro* studies of tumor invasion, Fig. 5g and h, the quantification shows that the cell number continues to increase over time in the other chambers (S1-4). Is this due to proliferation or influx of cells from S0? In which case, the hydrogel GNR is not inhibiting? It would be interesting to see other hydrogels as control in this experiment to gauge the potency of the test material.

Response: Thanks for your constructive suggestion. We are sorry for confusing you. Actually, the ranges of y-axis between **Fig.7c** (S0) and **Fig.7d** (S1-S4) in the revised manuscript corresponding to **Fig. 5g,h** in the initially submitted manuscript are different. In detail, the number of cells in S0 (**Fig.7c**) is above $7 \times 10^5/\text{mL}$, while in S1-S4 (**Fig.7d**), the number of cells is less than **20/mL** that is negligible comparing to that in S0, even though the incubation time exceeds 40 h. Thus, the increase of cell number over time was attributed to the proliferation, and hydrogel GNR still exerted a significantly inhibitory effect against cell migration. Related discussion has been highlighted in line 17 on page 18 to line 3 on page 19 of the revised manuscript.

According to the reviewer's suggestion, another control group, *i.e.*, pNIPAAm hydrogel that was not provided in the initial submission has been added in **Figs 1b, 5 and 6** of the revised manuscript. Since pNIPAAm features the lowest critical transition temperature of 32 °C, in the

absence of mild heating by laser irradiation, spontaneous gelation can occur. Since pNIPAAm features macropores and poor mechanical strength that have been demonstrated in **Fig. S13** of the revised supporting information, the internal stress arising from gelation fails to occlude blood vessels in tumor, as evidenced in **Figs S14 and S15** of the revised supporting information, which thus fails to effectively starve tumor (**Figs 5** of the revised manuscript). This result further validates the feasibility of hydrogel-GNR to enable starvation therapy *via* blood vessel embolism and blood & nutrition occlusion arising from the robust internal stress that gelation shrinkage causes. Related discussion has been added in line 20 on page 13 to line 12 on page 14 of the revised manuscript.

2 - Figure resolutions, especially where images are shown such as in Fig. 4d-h, is very low. Please provide higher magnification and higher resolution images for the reviewers.

Response: Thanks for your constructive suggestion. The poor image quality is probably attributed to the spontaneous compression when converting word-type manuscript into pdf-type manuscript during the online submission. To address it, another figure file including Figures 1-8 has been uploaded individually for reviewers, and the images in **Fig. 4d-h** in the initially submitted manuscript have been magnified so as to acquire higher magnification and resolution in **Fig. 5d-i** of the revised manuscript.

3 - Please provide an insight into the cells that are TUNEL positive in Fig.4d. Is it primarily the cancer cells or do the host cells also die? I suggest co-staining with a pancreatic epithelial cell marker at the minimum. This is important given the importance of the host cells in the tumor microenvironment for promoting primary tumor recurrence and metastasis. Also, I would like to see a syngeneic model in immune competent mice to get a sense of how robust the treatment is in overcoming the inflammatory milieu relevant to the solid tumor.

Response: Thanks for your constructive suggestion, which is highly appreciated. The images co-stained by FITC-labeled TUNEL, TRITC-labeled CD34 and DAPI-labeled nuclei in **Fig. 5g** of the revised manuscript have been provided to replace **Fig.4d** in the initially submitted manuscript for determining apoptosis source from host cells (endothelial cells) or cancer cells according to the reviewer's constructive suggestion. Semi-quantitative analysis has also been provided in **Fig. S17** of the revised supporting information. In detail, the Laser+hydrogel-GNR treatment achieves the

largest number of apoptosis including tumor cells and vascular endothelial cells, which is the reason that results in the delayed tumor growth. More significantly, the drastic decreases of CD34-labeled mature endothelial cells and CD31-labeled newborn endothelial cells after Laser+hydrogel-GNR treatment due to starvation-induced apoptosis indeed demonstrate the considerably reduced vascular density and anti-angiogenesis, as evidenced in **Fig. 5f,g** of the revised manuscript. As well, down-regulations of CD31&CD34 proteins representing vascular endothelial cells and up-regulations of pro-apoptotic proteins (*i.e.*, P53 and Caspase 3) representing cell apoptosis further demonstrate that Laser+hydrogel-GNR treatment causes the most apoptotic cells and tremendously decreases CD34 and CD31-labeled endothelial cells (that is, lowest blood vessel density), as shown in **Fig. 5i** of the revised manuscript. The decreased vascular density theoretically disfavors primary tumor recurrence and metastasis, which, thus, is expected to contribute to the inhibited tumor metastasis and recurrence along with the occluded migration pathway and occluded nutrition supply. Related discussion has been added in line 15 on page 14 to line 17 on page 16 of the revised manuscript.

According to the reviewer's suggestion, a syngeneic model, *i.e.*, BALB/c mice bearing 4T1 xenografted breast carcinoma, has been used as the immune competent mice to explore how robust the treatment is in overcoming the inflammatory milieu relevant to the solid tumor. More significantly, the influences of implants on inflammatory milieu relevant to the solid tumor have been tested. It has been found that the Laser+hydrogel-GNR treatment can reduce infiltrations of macrophages, dendritic (DC) cells and lymphocytes into tumor *via* squeezing and narrowing blood vessels and occluding blood & nutrition supply, and simultaneously exerts a robust inhibitory effect on inflammatory milieu. All experimental data have been provided in **Fig. 6f,g** of the revised manuscript. Related discussion has been provided in line 19 on page 17 to line 6 on page 18 of the revised manuscript.

In addition, evaluations on the gelation shrinkage-derived internal stress for tumor starvation therapy have been carried out on this syngeneic model, *i.e.*, BALB/c mice bearing 4T1 xenografted breast carcinoma. Firstly, the underlying principle, *i.e.*, internal stress squeezing for vascular embolization and blood & nutrition occlusion, has been also evaluated through using FITC-labeled dextrans with varied sizes, *i.e.*, 4K, 20K and 70K and clinically-used SonovueTM microbubbles to be detected by fluorescence imaging and contrast-enhanced harmonic imaging (CHI), respectively.

It has been found that the starvation therapy enabled by the vascular occlusion arising from the gelation shrinkage-derived internal stress successfully causes vascular embolism and occludes blood & nutrition supply, as demonstrated in **Fig. 6d,e** of the revised manuscript and **Supplementary Video S10-S13** of the supplementary videos, which, consequently, inhibits the tumor growth, as shown in **Fig.6a-c** of the revised manuscript. This result is similar to that in PANC-1 pancreatic xenografted tumor model, both of which demonstrate the ability of such a special starvation therapy to suppress tumor. Related discussion has been provided in line 18 on page 16 to line 18 on page 17 of the revised manuscript.

4 - The authors show ex vivo using intact blood vessel that the gel is able to block flow. How does this translate *in vivo*, especially in the tumors where the vasculature is leaky? I suggest *in vivo* perfusion studies with labeled dextran of varying sizes to test this. Would delivery of dextran to the tumors be limited due to constricted vessels? This would be an important and relevant study to do.

Response: Thanks for your kind suggestion. According to the reviewer's suggestion, objective to PANC-1 model, *in vivo* perfusion studies in PANC-1 pancreatic xenografted tumor model using FITC-labeled dextrans with varied sizes, *i.e.*, 4K, 20K and 70K, have been carried out, as shown in **Fig. 4b** of this revised manuscript. Additionally, *in vivo* perfusion studies with clinically-used SonovueTM microbubbles that can be detected by ultrasound contrast imaging under contrast-enhanced harmonic imaging (CHI) modality was also performed to demonstrate the constricted vessels by the gelation shrinkage-mediated squeezing in tumor, as shown in **Fig. 4c** of the revised manuscript and **Supplementary Videos S6-S9** of the supplementary videos. Both *in vivo* perfusion studies demonstrate that the constricted blood vessels successfully impede deliveries of dextrans and SonovueTM microbubbles due to vascular narrowing and embolism and blood occlusion caused by squeezing of the extravascular gelation shrinkage-derived internal stress in tumor. Related discussion has been added in line 3 on page 11 to line 2 on page 12 of the revised manuscript. **As well, direct vascular stenosis and narrowing caused by squeezing of gelation shrinkage-derived internal stress and intratumoral blood supply inhibition have been observed by CD34 and CD31 immunofluorescence staining in Fig. 4d,e of the revised manuscript.**

5 - Related to apoptosis that is evident in tumors: Fig. 4B shows a cytostatic effect of treatment and not a cytotoxic effect. How does this tie in with the apoptosis data? Another reason why it is important to identify which cells are undergoing apoptosis (percent of tumor and cell type), since it appears that this level is not sufficient to shrink the tumor.

Response: Thanks for your comments, which is highly appreciated. Measuring the variation of tumor volume as a function of time is the most common method that evaluates the therapeutic outcome of some therapeutic approaches and protocols. It has been extensively accepted that many therapeutic means failed to completely remove tumor and the residual tumor cells would re-grow, *e.g.*, radiotherapy and chemotherapy, which also exhibited a delayed growth phenomenon. However, this delayed process did not deny the presence of cytotoxic effect. Similarly, although Laser+hydrogel-GNR (2) treatment in **Fig. 5b** of the revised manuscript shows a cytostatic effect against tumor volume and receives a re-growth phenomenon after 10 days, the cytotoxic effect still contributes to the cytostatic tumor growth. In detail, TUNNEL immunofluorescence staining of tumor slices demonstrates that the treatment of Laser+hydrogel-GNR (2) brings about a large number of apoptotic cells, as evidenced in **Fig. 5g** of the revised manuscript, which directly demonstrates the presence of cytotoxic effect and contributes to the cytostatic effect against tumor volume. In addition, western blot data in **Fig. 5i** of this revised manuscript reflects the up-regulations of pro-apoptotic proteins (*e.g.*, P53, Caspase 3), which indirectly demonstrate the presence of cytotoxic effect. **Moreover, the HIF1 α up-regulation in Fig. 5h of this revised manuscript suggests that extravascular internal stress-mediated vascular narrowing and blood supply occlusion bring about hypoxia and induce apoptosis, which, along with occluded nutrition supply essential for tumor growth, ultimately delays the tumor growth (Fig. 5b).** The extravascular internal stress-mediated blood supply occlusion has been sufficiently demonstrated in *in vitro*, *ex vivo* and *in vivo* levels, as shown in **Figs 3 and 4** of the revised manuscript. Noticeably, the re-growth was probably attributed to stress recession arising from hydrogel degradation, as indicated in **Figs S26-S28** of the revised manuscript. Related discussion has been provided in line 1 on page 13 to line 17 on page 16 of the revised manuscript.

According to the reviewer's suggestion, semi-qualitative analysis according to TUNNEL immunofluorescence staining data (**Fig.5g**) was carried out to determine the apoptotic percentages of tumor cells and endothelial cells, as shown in **Fig. S17** of the revised supporting information

wherein the percentage of apoptotic cells after treatment with Laser+hydrogel-GNR (2) is 62% *via* calculating the ratio of green-labeled cells in whole blue-labeled cells. Furthermore, in apoptotic cells, CD34-expressing endothelial cells occupies over 18% *via* calculating the ratio of red and green co-labeled cells in whole green-labeled cells and approximately 80% apoptotic cells are tumor cells. Related discussion has been provided in line 2 to 7 on page 15 of the revised manuscript.

6 - A conceptual question that should be addressed in the discussion: Since the tumor is present after treatment, secondary treatment may be required such as chemotherapy. However, how do the authors suggest delivery of the chemotoxic agent if the gel-based treatment constricts the vasculature? At what stage of tumor development would this treatment be most effective? For early stage tumors surgical removal is still the accepted strategy. For late stage tumors this therapy may not be potent enough to act in a timely manner. Another concern is since the *in vivo* life is about 60 days. Will repeat procedures be required? This could counteract the anti-metastatic effect of the treatment. Looking at additional tumor models will increase the robustness by shedding light on applicability of treatment in a relevant context.

Response: Thanks for your constructive questions. We are very appreciated for your comprehensive and forward-looking suggestions that are beneficial for improving our manuscript. Actually, hydrogel as a drug reservoir has been extensively documented, among which chemotherapy or other therapeutic agents (*e.g.*, drug, siRNA, immune checkpoint inhibitors) can be gradually released in tumor when the gelation process occur (*e.g.*, *Nat Nanotechnol* **2019**, *14*, 89–97; *Nat Rev Mater* **2017**, *1*, 16071; *Sci Transl Med* **2018**, *10*, eaar1916; *Proc Natl Acad Sci U S A* **2017**, *114*, 5912-5917; *Nat Rev Drug Discovery* **2015**, *14*, 678-679; *Nat Nanotechnol* **2016**, *11*, 95-102; *et al.*), realizing chemotherapy. Thus, the secondary treatment, *i.e.*, chemotherapy, is expected *via* injecting anti-tumor drugs along with hydrogels. Related discussion has been added in line 5 to 14 on page 23 of the revised manuscript.

This fundamental research presents a new therapeutic strategy that enables starvation therapy *via* cutting off blood and nutrition supply by gelation shrinkage-mediated extravascular internal stress. Thus, the typical subcutaneous xenografted tumor on mice was used as the model to sufficiently demonstrate the feasibility of this strategy in this work. Although a systematic

evaluation on late-stage tumor treatment has not been carried out in this manuscript, **a preliminary evaluation using this special treatment method to inhibit metastasis of late tumors was enforced on the established transgenic mice model-bearing breast carcinoma that facily develop secondary metastatic tumors in lung**. Besides successfully inhibiting tumor metastasis, this special treatment in the treated group realizes a delayed growth for late-stage tumor in comparison to control group wherein metastasis occurs, as demonstrated in **Fig. 7g,h** of the revised manuscript and **Fig. S20** of the revised supporting information. Thus, it is still difficult currently to decide that at what stage of tumor development, this treatment would be mostly effective. Related discussion has been added in line 4 to 7 on page 24 of the revised manuscript.

Actually, residual tumor cells after surgery probably result in tumor recurrence or metastasis, and this special starvation strategy can serve as an auxiliary method to address the issues *via* reducing vascular density, cutting off blood & nutrition supply and occluding the migration passage of tumor cells. In detail, the Laser+hydrogel-GNR treatment can be applied to tumor periphery for inhibiting tumor recurrence and metastasis *via* cutting off nutrition supply essential for residual tumor and occluding their migration pathway, respectively, as demonstrated in **Figs 7 and 8** of this revised manuscript and **Fig. S21** of the revised supporting information. In the same way, the occluded migration pathway objective to late tumor is expected to inhibit metastasis. Related discussion has been provided in line 15 on page 23 to line 3 on page 24 of the revised manuscript.

The hydrogel is not robust and gradually degraded, as evidenced in **Figs S26-S28** of the revised supporting information. This process can progressively impair the internal stress, causing the gradual recovery of blood and nutrition supply, which is the reason why re-growth occurred after 10 days. To achieve a long-term anti-tumor outcome, repeated procedure is necessary. However, in this manuscript, the focus is to demonstrate the feasibility of our established starvation therapy. Thus, once treatment is sufficient and repeated treatment would influence the validation of this starvation therapy method, since the initially undegraded gelation probably influences the gelation of the next treatment, which determines that repeated treatment is not suggested. Instead, other treatment methods, *i.e.*, chemotherapy, immune therapy, *etc.*, can be potentially combined through entrapping anti-tumor drugs or immune checkpoint inhibitors in this composite hydrogel.

According to the reviewer's suggestion, another tumor model, *i.e.*, 4T1 carcinoma cancer subcutaneously implanted on BLAB/c mice has been used and this special treatment method has

been demonstrated to successfully starve tumor cells to death and realize the delayed tumor growth of 4T1 tumor *via* occluding blood & nutrition supply, as shown in **Fig. 6** of the revised manuscript, which sheds light on the applicability of this special treatment, Related discussion has been provided in line 18 on page 16 to line 6 on page 18 of the revised manuscript.

Sincerely thanks again for your comprehensive and insightful comments and suggestions. Enlightened by the reviewer's comments, future study will engage in the following three aspects, *i.e.*, implementing synergistic therapy combining this starvation therapy with chemotherapy, designing robust hydrogels capable of withstanding long erosion and sustaining stable internal stress, and carrying out late-stage tumor treatment using this special starvation strategy.

With a bit of additional data and more clarity, I recommend reconsideration for positive action and acceptance to Nature Communications.

Response: Thank you very much for your recommendation. We have tried our best to revise the manuscript according to your kind and construction comments and suggestions. We sincerely hope that this revised manuscript has addressed all your comments and suggestions.

Response to Reviewer #2

Reviewer #2 (Remarks to the Author):

This paper describes a gold nanorod and thermally-sensitive hydrogel mixture that undergoes gelation shrinkage. The authors use this gel to enable vascular shutdown on injection into tumor. The hypothesis is that the gelation causes the constriction of the blood supply to the tumor, which then results in suppression of metastasis and recurrence. This is a simplistic theory, but there are many questions.

Response: Thank you very much for your constructive comments and suggestions, which are highly appreciated. Please find the following detailed point-to-point responses to your comments and suggestions.

1. The gel is being injected into the tumor and allowed to stiffen. Although the authors perform ex vivo studies to show that this gelation blocks flow through a vessel, this has no relevance to a tumor microenvironment. For example, a major challenge in the tumor environment is diffusibility- does the gel diffuse uniformly through the tumor (could be tested using antibodies raised against the polymer or using tracers). A drop in blood flow in the tumor could just have been due to increased intratumoral pressure as a result of the injection of the hydrogel in the tumor.

Response: Thanks for your constructive suggestion. To demonstrate the diffusibility, according to the reviewer's kind reminding and suggestion, *in vivo* MRI and fluorescence imaging were carried out, as evidenced in **Fig. 4f** of the revised manuscript and **Fig. S10** of the revised supporting information, wherein Cy3-labeled dextran (70K) and Fe₃O₄ nanoparticles (400 nm) instead GNRs were used as tracers, respectively. Both experiments demonstrate that the rapid diffusion causes tracers to pervade the whole tumor within 1 min. It has been reported that 500 nm-sized silica nanoparticles after intratumoral administration can rapidly diffuse all over the whole tumor (*ACS Nano* **2013**, *7*, 6367-6377). Related discussion has been provided in line 12 to 19 on page 12 of the revised manuscript.

In normal abdomen, evident vascular narrowing arising from this special extravascular gelation

shrinkage-derived internal stress's squeezing has been successfully observed, as shown in **Fig. 3h** of the revised manuscript and **Supplementary Video S3** of the Supplementary Videos. To demonstrate the constricted vessels arising from the gelation shrinkage-mediated squeezing of blood vessels in tumor responsible for the drop in blood flow in the tumor rather than increased intratumoral pressure as a result of the injection of hydrogel in tumor, *in vivo* perfusion studies with FITC-labeled dextrans with varied sizes (*i.e.*, 4K, 20K and 70K) have been carried out. Related data have been supplemented in **Fig. 4b** of this revised manuscript wherein these FITC-labeled dextrans in treated group (*i.e.*, Laser+hydrogel-GNR) failed to enter the tumor in comparison to that in control group (*i.e.*, Laser+GNR). This result sufficiently demonstrates that the vascular narrowing and occlusion are responsible for the reduced blood velocity and occluded blood supply rather than the increased intratumoral pressure as a result of the injection of hydrogel in tumor. Related discussion has been provided in line 3 to 13 on page 11 of the revised manuscript.

Moreover, *in vivo* perfusion studies with clinically-used Sonovue™ microbubbles that can be detected by ultrasound contrast imaging under contrast-enhanced harmonic imaging (CHI) modality was further performed to demonstrate that the constricted vessels' narrowing arising from the gelation shrinkage internal stress-mediated squeezing of blood vessels in tumor is responsible for the drop in blood flow, as shown in **Fig. 4c** of this revised manuscript and **Supplementary Videos S6-S9** of this supplementary videos. Related discussion has been added in line 14 on page 11 to line 2 on page 12 of the revised manuscript.

2. What is strange is the reduction in biomarkers of vasculature following injection. If the mechanism is *via* occlusion, the biomarker levels should not change. The levels should only change if an agent that prevents angiogenesis (or causes the destruction of endothelial cells) is administered. Histology images in Fig.4 are not convincing of the claims made.

Response: Thanks very much for your comment, which is highly appreciated. After short period (less than 12 h) post-corresponding treatment, only vascular narrowing was observed and no reduced vascular density was obtained, as evidenced in **Fig. 4d,e** of the revised manuscript. In contrast, the pathological examinations were carried out after 10 days post-treatment that was the optimal time point when the inhibitory effect reached the best. Due to the long period of blood & nutrition occlusion induced by the squeezing and narrowing that extravascular gelation

shrinkage-derived internal stress caused, hypoxia and apoptosis were induced, as confirmed by the up-regulations of hypoxia inducible factor 1 α (HIF1 α) and pro-apoptotic protein (P53) (**Fig. 5h,i**). **Thus, it is not difficult to understand that the long-term deficiency of nutrition and oxygen essential for tumor cells and vascular endothelial cells inevitably results in cell apoptosis including tumor cells and CD34-labeled endothelial cells and reduce vascular density, as evidenced by TUNEL immunofluorescence staining with CD34 co-staining in Fig. 5g of this revised manuscript and Fig. S17 of the revised supporting information.** The down-regulation of CD34 protein also reflects the reduced vascular density (**Fig. 5i** of the revised manuscript). Moreover, the inhibited nutrition supply also inhibits CD31-labeled angiogenesis and PCNA-labeled cell proliferation due to lack of energy, nutrition and oxygen supply for cell proliferation, as confirmed by PCNA (**Fig. 5e**) and CD31 (**Fig. 5f**) immunohistochemical staining. As well, western blot data also demonstrate the down-regulation of CD31 protein. Related discussion has been provided in line 15 on page 14 to line 17 on page 16 of the revised manuscript.

3. One should have expected occlusion to increase HIF1 α . Instead, the authors claim an increase in P53. How is that happening?

Response: Thanks very much for your constructive question. Analysis of HIF1 α *via* western blot has been provided in the revised manuscript. As expected, the up-regulation of HIF1 α is observed in **Fig. 5h** of the revised manuscript.

As stated in response to comment 2, this treatment strategy can cease blood and nutrition supply *via* blood occlusion arising from the extravascular gelation shrinkage-derived internal stress. The long-term vascular narrowing or embolism will induce hypoxia and cause HIF1 α elevation (as indicated in **Fig. 5h** of the revised manuscript), and further induce cell apoptosis including tumor cells and vascular endothelial cells, as demonstrated by increased apoptotic cells in TUNEL immunofluorescence staining (**Fig. 5g** of the revised manuscript) and semi- quantitative analysis (**Fig. S17** of the revised supporting information) and pro-apoptotic protein up-regulation (*i.e.*, P53) in western blot (**Fig. 5i**) of this revised manuscript. The similar phenomenon has also been reported by Shi et al group (*Nat Nanotechnol* **2017**, *12*, 378–386) who introduced an intravascular aggregates-mediated blood & nutrition occlusion for starvation therapy and found HIF1 α elevation and P53 elevation. It has been documented that starvation therapy can elevate P53 expression

(*EMBO J* **2007**, 26, 4812–4823) and induce hypoxia (*Biomaterials* **2018**, 162, 123-131; *Acc Chem Res* **2018**, 51, 2502-2511; *Mol Cancer* **2015**, 14, 77; *et al.*). Furthermore, many references has reported that hypoxia-induced HIF1 elevation can trigger apoptosis through the stabilization of P53 and high expressions of both HIF-1 α and p53 are essential for the hypoxia-induced cell death (*e.g.*, *Cell Death Dis* **2011**, 2, e164; *Nature* **1998**, 392, 405-408; *Biochem Biophys Res Commun* **2005**, 331, 718–725; *Nat Rev Cancer* **2003**, 3, 721-732; *Mol Cell Biochem* **2016**, 415, 29–38; *J Clin Pathol* **2004**, 57, 1009–1014; *Biochem Pharmacol* **2002**, 64, 889-892; *et al.*). **Thus, these previous researches provide robust support for our above experimental results that the blood vessel occlusion-induced hypoxia indeed promotes the up-regulations of HIF1 α and P53 and induces cell apoptosis including tumor cells and endothelial cells.** The related discussion has been provided in line 11 on page 15 to line 17 on page 16 of the revised manuscript

4. In Fig. 5, the tumor is shown to metastasize from one subcutaneous site of injection to another subcutaneous site. The authors claim these sub-cut sites as distant metastasis sites. In my 25 years of working in cancer biology, I am yet to see such subcutaneous to subcutaneous metastasis. Are the authors sure that the cells were not injected into the other subcutaneous sites by mistake? Typically, one would see met to distant organs.

Response: Thanks for your constructive suggestion. The subcutaneous to subcutaneous metastasis model *via* microbubbles-assisted ultrasound-enhanced vascular destruction is accessible, and more detailed investigation is ongoing. To remove the reviewer's doubt, a transgenic mouse model that can generate mammary adenocarcinomas after induction by MMTV-PyVmT oncogene has been accepted and used as the ideal model for evaluating anti-metastasis ability of this special treatment in the revised manuscript, as the majority of the tumor-bearing transgenic mice develop secondary metastatic tumors in the lung, as evidenced by many references (*e.g.*, *Cancer Res* **2007**, 67, 3106-3116; *Oncol Res* **2017**, 25, 407-415; *Mol Cell Biol* **1992**, 12, 954–961, *et al.*). It has been found that the Laser+hydrogel-GNR treatment (treated group) can effectively inhibit lung metastasis from the primary mammary tumor comparing to the control group, as shown in **Fig. 7e-h** of the revised manuscript. Related discussion has been added in line 12 on page 19 to line 6 on page 20 of the revised manuscript.

5. The authors claim that this approach avoids intravascular aggregates-induced embolisms. Given that the hydrogel disappears after a period of time, one needs to test where it goes, and degradation products in blood could have limitations as current intravascular approaches. If the hydrogel degrades over time, why is recurrence inhibited?

Response: Thanks for your insightful comment and question. Identical to all other organic carriers, the *in vivo* degradation products and their fates of pNIPAAm-based hydrogels have failed to be directly determined, and almost all reported researches only focus on *in vitro* determination of degradation products based on the difficulty for *in vivo* evaluation on the biodegradation process and products, *e.g.*, *Pharm Res* **2014**, *31*, 742–753; *J Biomed Mater Res Part A* **2008**, *87A*, 345–358; *Adv Funct Mater* **2017**, *27*, 1704107; *et al.* The reason for this phenomenon lies in the fact that the composition atoms (*i.e.*, C, H, O, N) of pNIPAAm-based hydrogels are ubiquitous in all organs and metabolism products of living body and no specially-labeled atom is nonexistent in living body, which determines the inapplicability of many test methods with atom analysis as the underlying principle. As well, the degradation products (*i.e.*, pNIPAAm segments) feature different molecular weights and different metabolism products in living body also share different sizes, which determine the inapplicability of many test methods with molecular weight or size difference as the underlying principle. Currently, in *in vivo* level, the degradation rate of implants can be obtained according to the relative volume variation of embedded implant (*ACS Appl. Mater. Interfaces* **2009**, *1*, 319–327). **Actually, in this manuscript, the composite hydrogel (*i.e.*, hydrogel–GNR) has been demonstrated to gradually degrade *via* measuring the relative volume variation of embedded hydrogel-GNR, as shown in Fig. S27 of the revised supporting information.** Related discussion has been provided in line 9 to 20 on page 22 of the revised manuscript.

Although no direct method was used to detect the fates of degradation products of pNIPAAm-based hydrogels, the fates of GNRs that are chelated to the hydrogel skeleton and label this composite hydrogel were detected to indirectly reflect the fates of degradation products (*i.e.*, pNIPAAm segments) of pNIPAAm-based hydrogels, since the covalent bonding of GNRs with hydrogel skeleton is robust enough to withstand erosion or degradation and Au atom is almost inexistent in living body. This indirect method is similar to radio-labeled test method (*Biomaterials* **2009**, *30*, 2598–2605). As shown in **Fig. S28** of the revised supporting information, after 60 days, 80% Au atoms are excreted through urine and feces, indirectly

suggesting that the excretion pathway of most degradation products of such a composite hydrogel is urine and feces. Related discussion has been provided in line 20 on page 22 to line 3 on page 23 of the revised manuscript.

It has been extensively accepted that pNIPAAm-based hydrogel and their degradation products feature excellent biosafety and they have been widely used, *e.g.*, *Nature* **1997**, 388, 860-862; *Adv Mater* **2013**, 25, 6737-6743; *Acta Biomater* **2016**, 32, 10-23; *Pharm Res* **2014**, 31, 742-753; *J Biomed Mater Res Part A* **2008**, 87A, 345-358; *Adv Funct Mater* **2017**, 27, 1704107; *et al.* Moreover, the biosafety experiments including blood and biochemical tests and pathological examinations (*i.e.*, H&E staining) of normal tissues demonstrate the excellent biosafety of this composite hydrogel, and no embolism is observed in the H&E pathological staining, as evidenced in **Figs S23-S25** of the revised supporting information. Related discussion has been provided in line 1 to 8 on page 22 of the revised manuscript.

As for inhibited recurrence, the marriage of laser irradiation with hydrogel-GNR that was injected into the tumor periphery served as an assisted method to avoid recurrence after photothermal ablation. After photothermal ablation, residual tumor cells were starved to death before invalidation of adjacent blood vessels occlusion by this gelation shrinkage-derived internal stress due to the gradual degradation of gels. **Thus, in anti-recurrence, the sole question is to design robust hydrogels with appropriate degradability rate that guarantees continuous work for occluding blood & nutrition supply and starving the residual tumor cells to death before their failures.** Taken all together, this gelation shrinkage-derived internal stress strategy is equipped with starvation therapy, anti-metastasis and anti-recurrence as an assistance of photothermal ablation or other treatment methods. **Future studies will engage in the following three aspects, *i.e.*, implementing synergistic therapy combining this starvation therapy with chemotherapy, designing robust hydrogels capable of withstanding long erosion and sustaining stable internal stress, and carrying out late-stage tumor treatment using this special starvation strategy.** Related discussion has also been provided in line 15 on page 23 to line 10 on page 24 of the revised manuscript.

6. Minor points:

a. Please use exact numbers instead of 'some mice were sacrificed'. Provide n values for every studies

in legend, together with statistical tests performed.

b. In some sentences, parts are missing; for example, line 102- I presume LCST of the pure polymer is 32C.

Response: Thanks very much for your constructive suggestion and kind reminding, which is highly appreciated. We have scrutinized the manuscript, revised the typos and provided the missed details in the corresponding places of the revised manuscript and the revised supporting information accordingly to the reviewer's suggestion.

Reviewers' comments:

Reviewer #1 (Remarks to the Author):

First, I want to commend the authors on addressing reviewers' comments by including additional data from two new animal tumor models (MMTV-PyMt and 4T1 syngeneic), intra-tumoral assessment of vessel occlusion, evaluation of inflammation within the tumor (immunophenotyping and cytokines) and hypoxia (Hif1a). Through these studies the authors have tried to address key questions such as generality of treatment and in vivo feasibility. However, I want to draw their attention to key portions of manuscript that are unclear.

- 1) In Fig. 4b, it appears that the treated mice has more signal from 70K. This does not fit the description in text or hypothesis.
- 2) Similarly, in Fig. 4e, the treatment group shows more CD31 staining. Please check if this is correct.
- 3) Fig. 4d and 4e, IF for CD34 and CD31 demonstrate "abundance" of vessels, not "narrowing". Please re-state in text.
- 4) In general, data from various imaging techniques forms a bulk of the results. However, all the imaging data is qualitative. Please provide quantification for the imaging data (perfusion exps., IF etc.) to make a strong objective point. For instance, I would place the supplemental figure S17 in the main text.
- 5) In supplemental figure S18, are the differences statistically different? The blots do not appear to be strikingly different except for Hif1.
- 6) IHC in figures 5e and 5f for the control (positive) is really faint. Please replace with better stained images.
- 7) Fig. 5g: At the present magnification, it is very difficult to see co-localization. Please update with a higher magnification to show clear co-localization. It can be accompanied by quantification from a larger field of review to be truly representative.
- 8) While I am glad the authors look into the key phenotype of inflammation, the data is not clearly presented. In fig. 6f, which cytokines are differentially regulated by treatment? Please separate out into individual graphs, show statistics on the graph, and maybe include some in the supplemental as opposed to the main text.
- 9) Similarly, please show data quantification for immunophenotyping in fig. 6g as the dot plots are difficult to read. Dot plots should be included in the supplemental but may be removed from the main text.

10) Please explain the methodology for tumor recurrence study. I am unclear as to why there was repeat GNR injection in this study for both control and treatment arms. How long after treatment was tumor recurrence checked (days, weeks?)?

11) Please have the manuscript reviewed for language. As an example, the first sentence of Discussion needs editing. There are a few typos as well, for instance PABC-1 and BLAB/c in the discussion.

Please consider after minor revisions for language and figure modifications.

Reviewer #2 (Remarks to the Author):

The authors have done substantial experiments to address the questions. While the findings are interesting, the rigor is weak. However, the way the results are presented fails to convince this reviewer. For example,

1. in Fig. 4B, not sure what the differences are. The authors say that the FITC-Dextran is not allowed to enter the treated tumors, but its hard to confirm from the images. The same applies to Fig. 6.

2. Similarly, all IHCs need to be stereologically analyzed and quantified. All Western blots should similarly be quantified. Please provide the number of times the experiments were repeated to get the quantification.

3. The authors claim that the treatments have different effect on cytokine levels- Fig.6F. There is no statistical analysis performed. From just eyeballing, the results do not seem to support what the authors claim.

While the concept is interesting, the lack of rigor weakens the manuscript. Furthermore, the authors can benefit from a tighter and better scientific writing.

Response to Reviewer #1

Reviewer #1 (Remarks to the Author):

First, I want to commend the authors on addressing reviewers' comments by including additional data from two new animal tumor models (MMTV-PyMt and 4T1 syngeneic), intra-tumoral assessment of vessel occlusion, evaluation of inflammation within the tumor (immunophenotyping and cytokines) and hypoxia (Hif1a). Through these studies the authors have tried to address key questions such as generality of treatment and *in vivo* feasibility. However, I want to draw their attention to key portions of manuscript that are unclear.

Response: Thank you very much for the positive comments and constructive suggestions.

Please find the following detailed responses to your comments and suggestions.

1) In Fig. 4b, it appears that the treated mice have more signal from 70K. This does not fit the description in text or hypothesis.

Response: Thanks very much for your comment, which is highly appreciated. In Fig. 4b, the scale bars in 70 K, 20 K, and 4 K are different. More significantly, different sizes caused different distributions of dextrans with different sizes in different organs of mice. Therefore, it is theoretically hard and meaningless to compare the signal intensity between two different dextrans-injected groups, because the aim of perfusion experiments using different FITC-labeled dextrans with variable sizes merely demonstrated the vascular narrowing-mediated blood occlusion. And, the aim is easily validated *via* comparing treated (*i.e.*, Laser+hydrogel-GNR) and control in either dextran-treated group.

In this regard, we did not compare and describe the signal between either two different dextran-injected groups (*i.e.*, 70K vs 4K, 70K vs 20K or 20K vs 4K) in the main text. The tumor indicated by dotted circle and arrow was underlined, as shown in Figure 1a of this response letter and Fig. 4b of the revised manuscript, wherein the intratumoral distributions of dextrans between treated and control in either dextran-injected group were compared, and the quantitative data were obtained in Figure 1b of this response letter and Fig. 4c of the revised manuscript. Related discussion has been added in line 3 to 7 on page 11 of the revised manuscript.

Figure 1. (a) Animal fluorescence images of mice in treated and control groups, wherein FITC-labeled dextrans with variable sizes (*i.e.*, 70K, 20K and 4K) were used as tracers to evaluate whether the treatment with in treated group can occlude intratumoral blood vessels and hamper the penetration of dextran into tumors in comparison to mice in control group. Note: the regions of interest indicate tumor that is marked by green dotted circle and arrow. (b) Signal intensities of FITC conjugated to dextran in tumor of interest indicated by dotted green circle and arrow in (a). Experiments were repeated three times. Data are expressed as mean \pm SD ($n = 3$). *** $P < 0.001$, and the statistical significances were obtained using t-student test.

2) Similarly, in Fig. 4e, the treatment group shows more CD31 staining. Please check if this is correct.

Response: Thanks very much for your constructive suggestion and kind reminding, which is highly appreciated. We are sorry for this confusing issue. The CD31 staining was instantly carried out without delay after treatment with Laser+hydrogel-GNR, thus the vascular density should not be altered, and more CD31 staining in treatment group is probably attributed to the varied parameter

of laser confocal scanning microscopy (LCSM) in comparison to control group due to our negligence. To address the confusion, a new group of images in the treatment group (Treated) has been updated to replace the old ones in Fig.4g of the revised manuscript and Figure 2b of this response letter.

Figure 2. Matured (a) and budding (b) blood vessels of PANC-1 xenografted pancreatic tumors after corresponding treatments in control and treated groups, and the two types of blood vessels were stained by FITC-labeled CD34 immunofluorescence (a) and Cy3-labeled CD31 immunofluorescence (b). Notes: yellow arrows indicate the blood vessels and in treated group vascular narrowing is clearly found in comparison to that in control group.

3) Fig. 4d and 4e, IF for CD34 and CD31 demonstrate “abundance” of vessels, not “narrowing”. Please re-state in text.

Response: Thanks for your constructive suggestion. The CD31 & CD34 stainings were instantly carried out without delay after treatment with laser+hydrogel-GNR, thus the vascular density would not be altered. The experiments aimed at validating the vascular narrowing arising from hydrogel

shrinkage-squeezing. Besides successfully confirming “abundance” of vessels, the vascular narrowing is also clearly observed, as indicated by the yellow arrows in Fig. 4f,g of the revised manuscript and Figure 2 of this response letter. Related discussion has been added in line 1 to 6 on page 12 of the revised manuscript.

4) In general, data from various imaging techniques forms a bulk of the results. However, all the imaging data is qualitative. Please provide quantification for the imaging data (perfusion exps., IF etc.) to make a strong objective point. For instance, I would place the supplemental figure S17 in the main text.

Response: Thanks for your constructive suggestion, which is highly appreciated. Quantification for the imaging data including perfusion experiments (*i.e.*, contrast-enhanced ultrasound imaging and FITC-labeled dextran-enhanced animal fluorescence imaging) and IF have been provided in appropriate combined figures, *e.g.*, Figs 4c, 4e, 6b, 6c, 7e and 7g in the revised manuscript. The Supplemental Figure S17 has moved to Fig. 6b,c in this revised manuscript according to the reviewer’s suggestion. Related depictions have been added in the according places in the revised manuscript.

5) In supplemental figure S18, are the differences statistically different? The blots do not appear to be strikingly different except for Hif1.

Response: Thanks very much for your constructive question. Statistical differences have been added in the supplemental Figure S18 of the revised supporting information.

6) IHC in figures 5e and 5f for the control (positive) is really faint. Please replace with better stained images.

Response: Thanks for your constructive suggestion. According to the reviewer’s suggestion, better stained images have been used to replace the old ones in Fig. 5e,f of the revised manuscript.

7) Fig. 5g: At the present magnification, it is very difficult to see co-localization. Please update with a higher magnification to show clear co-localization. It can be accompanied by quantification from a larger field of review to be truly representative.

Response: Thanks for your constructive suggestion, which is highly appreciated. According to the reviewer's suggestion, Fig. 5g in the original manuscript has been magnified and combined with the quantification graphs into an individual figure (*i.e.*, Figure 6) in this revised manuscript.

8) While I am glad the authors look into the key phenotype of inflammation, the data is not clearly presented. In fig. 6f, which cytokines are differentially regulated by treatment? Please separate out into individual graphs, show statistics on the graph, and maybe include some in the supplemental as opposed to the main text.

Response: Thanks very much for your constructive suggestion, which is highly appreciated. According to the reviewer's suggestion, Fig. 6f in the original manuscript has been divided into Fig. 7h in the revised manuscript and Supplementary Figure S20 in the revised supporting information, respectively. Accordingly, the statistics have been added appropriately in these graphs.

9) Similarly, please show data quantification for immunophenotyping in fig. 6g as the dot plots are difficult to read. Dot plots should be included in the supplemental but may be removed from the main text.

Response: Thanks for your kind suggestion. According to the reviewer's suggestion, data quantification for immunophenotyping in Fig. 6g of the original manuscript has been divided into Fig. 7i of the revised manuscript and Supplementary Figure S22 of the revised supporting information, respectively. Dot plots have been removed from the main text and moved to Figure S21 of the revised supporting information.

10) Please explain the methodology for tumor recurrence study. I am unclear as to why there was repeat GNR injection in this study for both control and treatment arms. How long after treatment was tumor recurrence checked (days, weeks)?

Response: Thanks for your constructive questions, which is highly appreciated. In the experiment regarding tumor recurrence study, hydrogel-GNR was firstly injected into the surrounding periphery of PANC-1 tumor and laser (1) irradiation was carried out, followed by GNRs injection into the PANC-1 tumor and laser (2) irradiation for ablating PANC-1 tumor. Herein, intratumoral injection of GNRs into PANC-1 tumor was expected to incompletely ablate tumor in the presence of laser (2)

irradiation and generate residual tumor tissues available for constructing recurrence model. However, the pre-treatment with injection of hydrogel-GNR into the normal tissues surrounding PANC-1 tumor and laser (1) irradiation could trigger gelation process of such a composite hydrogel, narrow periphery blood vessels surrounding tumor and occlude blood and nutrition supply towards residual tumor tissues, consequently suppressing tumor recurrence. In contrast, in control group, PBS instead of hydrogel-GNR was used, which failed to occlude blood and nutrition supply towards residual tumor tissues and allowed tumor recurrence.

After 5 days post-treatment, tumor recurrence was checked. Related explanation associated with experiment design has been provided in line 16 to 22 on page 38 of the revised supporting information.

11) Please have the manuscript reviewed for language. As an example, the first sentence of Discussion needs editing. There are a few typos as well, for instance PABC-1 and BLAB/c in the discussion.

Response: Thanks for your constructive suggestion. We have scrutinized the manuscript, and made according revisions including some typos, grammatical errors and long sentences, *etc.*, and polished the language in the revised manuscript.

Please consider after minor revisions for language and figure modifications.

Response: Thank you very much for your recommendation. We have tried our best to revise the manuscript according to your kind and construction comments and suggestions. We sincerely hope that this revised manuscript has addressed all your comments and suggestions.

Response to Reviewer #2

Reviewer #2 (Remarks to the Author):

The authors have done substantial experiments to address the questions. While the findings are interesting, the rigor is weak. However, the way the results are presented fails to convince this reviewer. For example,

1. in Fig. 4B, not sure what the differences are. The authors say that the FITC-Dextran is not allowed to enter the treated tumors, but its hard to confirm from the images. The same applies to Fig. 6.

Response: Thanks for your comments, which is highly appreciated. In Fig.4b of the revised manuscript or Figure 1a of the response letter, no matter which size of FITC-labeled dextrans was used, much stronger fluorescence signal in tumor that is indicated by green dotted circle and green arrow is observed in control group. In contrast, almost no fluorescence is obtained in treated group. Quantitatively, the signal intensity at the site of PANC-1 tumor in control group is far larger than that in treated group, as evidenced in Fig. 4c of the revised manuscript or Figure 1b of this response letter. This phenomenon sufficiently suggests that FITC-labeled dextrans were not allowed to enter the treated tumors due to vascular occlusion caused by gelation shrinkage in treated group. Related discussion has been provided in line 3 to 7 on page 11 of the revised manuscript.

Similar phenomenon occurred to the perfusion experiment on 4T1 model using FITC-labeled dextrans with varied sizes in Fig 7f,g of the revised manuscript and Figure 3 of this response letter. Related discussion has been provided in line 9 to 13 on page 17 of the revised manuscript.

Figure 3. *In vivo* animal fluorescence images (a) and quantitative signal intensities (b) of 4T1 xenografted tumor on BALB/c mice after corresponding treatments in control (*i.e.*, Laser+GNR) and treated (*i.e.*, Laser+hydrogel-GNR (2)) groups and subsequent injection of different FITC-labeled dextrans with varied sizes (*i.e.*, 70K, 20K and 4K), wherein yellow arrows and dotted circles indicate the tumor. Experiments were repeated three times. *** $P < 0.001$ and the statistical significances were obtained using t-student test. Data are expressed as mean \pm SD (n = 3).

2. Similarly, all IHCs need to be stereologically analyzed and quantified. All Western blots should similarly be quantified. Please provide the number of times the experiments were repeated to get the quantification.

Response: Thank you very much for your constructive suggestions, which are highly appreciated.

Data quantification on all IHCs has been provided in Supplement Figure S17 of the revised supporting information, and all Western blots in Fig. 5g of the revised manuscript have been also quantified in Supplement Figure S18 of the revised supporting information. Triple times were

repeated to get the quantification, and it has been added in the corresponding Figure captions.

3. The authors claim that the treatments have different effect on cytokine levels- Fig.6F. There is no statistical analysis performed. From just eyeballing, the results do not seem to support what the authors claim.

Response: Thank you very much for your constructive comments and suggestions, which are highly appreciated. Statistical analysis has been added in the graphs of cytokine secretion including Fig. 7h of the revised manuscript and Supplement Figure S20 of the revised supporting information, wherein the statistical differences of some cytokines in G4 (*i.e.*, Laser+hydrogel-GNR (2)) in comparison to G1 (*i.e.*, control group) were obtained.

While the concept is interesting, the lack of rigor weakens the manuscript. Furthermore, the authors can benefit from a tighter and better scientific writing.

Response: Thank you very much for your constructive comments. We have tried our best to revise the manuscript according to your kind and construction comments and suggestions including language polishing, revisions of typos, graphs or figures. Especially some missed quantitative data and statistical analysis in graphs have been supplemented in the revised manuscript. We sincerely hope that this revised manuscript has addressed all your comments and suggestions.

REVIEWERS' COMMENTS:

Reviewer #1 (Remarks to the Author):

The authors have responded appropriately to all reviewers requests. Publication is recommended.

Reviewer #2 (Remarks to the Author):

The authors have addressed my comments. I have no further comments.